# Gallium Nitrate Enhances Antimicrobial Activity of Colistin against *Klebsiella pneumoniae* by Inducing Reactive Oxygen Species Accumulation

Mingjuan Guo,[a] Ping Tian,[b] Qingqing Li,[b] Bao Meng,[b] Yuting Ding,[b] Yanyan Liu,[b,c,d] Yasheng Li,[b,c,d] Liang Yu,[b,c,d] Jiabin Li[a,b,c,d]

[a]Department of Infectious Disease, The Chaohu Affiliated Hospital of Anhui Medical University, Hefei, China
[b]Department of Infectious Disease, The First Affiliated Hospital of Anhui Medical University, Hefei, China
[c]Anhui Center for Surveillance of Bacterial Resistance, Hefei, China
[d]Institute of Bacterial Resistance, Anhui Medical University, Hefei, Anhui, China

Mingjuan Guo and Liang Yu contributed equally to this work. Author order was determined by drawing straws.

**ABSTRACT** *Klebsiella pneumoniae*, a pathogen of critical clinical concern, urgently demands effective therapeutic options owing to its drug resistance. Polymyxins are increasingly regarded as a last-line therapeutic option for the treatment of multidrug-resistant (MDR) Gram-negative bacterial infections. However, polymyxin resistance in *K. pneumoniae* is an emerging issue. Here, we report that gallium nitrate (GaNt), an antimicrobial candidate, exhibits a potentiating effect on colistin against MDR *K. pneumoniae* clinical isolates. To further confirm this, we investigated the efficacy of combined GaNt and colistin *in vitro* using spot dilution and rapid time-kill assays and growth curve inhibition tests and *in vivo* using a murine lung infection model. The results showed that GaNt significantly increased the antimicrobial activity of colistin, especially in the iron-limiting media. Mechanistic studies demonstrated that bacterial antioxidant activity was repressed by GaNt, as revealed by RNA sequencing (RNA-seq), leading to intracellular accumulation of reactive oxygen species (ROS) in *K. pneumoniae*, which was enhanced in the presence of colistin. Therefore, oxidative stress induced by GaNt and colistin augments the colistin-mediated killing of wild-type cells, which can be abolished by dimethyl sulfoxide (DMSO), an effective ROS scavenger. Collectively, our study indicates that GaNt has a notable impact on the antimicrobial activity of colistin against *K. pneumoniae*, revealing the potential of GaNt as a novel colistin adjuvant to improve the treatment outcomes of bacterial infections.

**IMPORTANCE** This study aimed to determine the antimicrobial activity of GaNt combined with colistin against *Klebsiella pneumoniae in vitro* and *in vivo*. Our results suggest that by combining GaNt with colistin, antioxidant activity was suppressed and reactive oxygen species accumulation was induced in bacterial cells, enhancing antimicrobial activity against *K. pneumoniae*. We found that GaNt functioned as an antibiotic adjuvant when combined with colistin by inhibiting the growth of multidrug-resistant *K. pneumoniae*. Our study provides insight into the use of an adjuvant to boost the antibiotic potential of colistin for treating infections caused by multidrug-resistant *K. pneumoniae*.

**KEYWORDS** gallium, colistin, *Klebsiella pneumoniae*, reactive oxygen species, bacterial antioxidant activity, oxidative stress

It is widely recognized that an increase in antibiotic resistance poses a serious threat to humanity. The World Health Organization (WHO) has identified *Enterobacteriaceae* as a top-priority pathogens for research and development of new therapeutics because these bacteria have become resistant to a large number of antibiotics, including carbapenems

Address correspondence to Liang Yu, yu2447438@gmail.com, or Jiabin Li, lijiabin@ahmu.edu.cn.

The authors declare no conflict of interest.

and third-generation cephalosporins (1, 2). *Klebsiella pneumoniae*, a member of the *Enterobacteriaceae* family, is associated with high morbidity and mortality. Moreover, multidrug-resistant *K. pneumoniae* has emerged worldwide, accounting for approximately 30% of all Gram-negative infections, including pneumonia, urinary tract infections, and bacteremia (3, 4). The combination of nosocomial spread and MDR *K. pneumoniae* strains is becoming a major challenge for clinicians, as few treatment options remain currently available (5).

Owing to increasing antibiotic resistance, the demand for novel antimicrobials targeting MDR *K. pneumoniae* is growing. Polymyxin antibiotics (including polymyxin B and colistin) are increasingly being considered as the last option for treating infections caused by MDR bacteria that are resistant to nearly all other currently available antibiotics (6, 7). Polymyxins can induce the formation of reactive oxygen species (ROS) via membrane lysis, eventually causing cell death (8, 9). However, transferable polymyxin resistance mediated by mobile colistin resistance (MCR) enzymes fueled the generation of polymyxin-resistant bacteria, including polymyxin-resistant *K. pneumoniae* (10–12). The increased frequency of polymyxin-resistant *K. pneumoniae* frequencies worldwide has resulted in *K. pneumoniae* infections that are very difficult to treat and are thus associated with higher mortality rates (13). An alternative strategy to alleviate this crisis is to identify promising compounds that can restore polymyxin activity (14).

Gallium is a naturally occurring metal with an ionic radius that is nearly identical to that of iron but is redox-inactive (15). Because gallium is a ferric iron mimetic, gallium compounds were used to disrupt bacterial iron metabolism and had antibacterial activity against several human pathogens, including *K. pneumoniae*, *Pseudomonas aeruginosa*, *Francisella tularensis*, *Acinetobacter baumannii*, *Staphylococcus aureus*, and several mycobacterial species, among others (16). Moreover, because gallium nitrate (GaNt) can substitute for iron in many proteins and interfere with multiple bacterial functions, the resistance rates toward GaNt are low in *Escherichia coli* (17) and *P. aeruginosa* (18). Therefore, GaNt, the FDA-approved GaNt formulation, represents a candidate for reinvestigation and has received considerable attention as a metal compound with broad-spectrum antimicrobial activity. Notably, recent studies have suggested that GaNt could induce the intracellular accumulation of ROS and transcription of genes involved in ROS detoxification in bacteria (19). Previous studies have also demonstrated that GaNt synergized with antibiotics against *P. aeruginosa* (18, 20) and potentiate the antibacterial effect of gentamicin against *F. tularensis* (21). However, the effect of combinatorial treatment on the growth and spread of antibiotic-resistant bacteria is still not fully understood.

In this study, we explored the combined activity of GaNt and colistin against *K. pneumoniae in vitro* and *in vivo*. We found that GaNt significantly increased intracellular ROS levels, thereby increasing the antimicrobial activity of colistin. The result was further confirmed by RNA-seq, and the expression of various genes involved in antioxidant activity was found to be suppressed by GaNt. Collectively, our data demonstrate the potential of GaNt as an adjuvant therapy for treating MDR bacterial infections.

## RESULTS

**Identification of MDR *K. pneumoniae* clinical isolates.** The 1,200 *K. pneumoniae* clinical isolates used in this study were obtained from the Anhui Center for Surveillance of Bacterial Resistance from 2017 to 2020. The MICs of these strains against 24 antibiotics were determined using the agar dilution method. According to the American CLSI guidelines, approximately 79.1% (*n* = 949) of the tested isolates had an MDR phenotype, 5.94% (*n* = 56) of which were resistant to colistin; 32.1% (*n* = 18) of the colistin (CST)-resistant isolates exhibited carbapenem-resistant phenotypes.

A total of 20 multidrug-resistant clinical isolates were randomly selected for follow-up studies, including 10 CST-resistant and 10 CST-sensitive strains (see Table S1 in the supplemental material). These strains were identified as *K. pneumoniae* using matrix-assisted laser desorption ionization–time of flight mass spectrometry (MALDI-TOF MS) (Fig. S1). Homology analysis of the 20 strains of *K. pneumoniae* was conducted using

## PFGE-XbaI

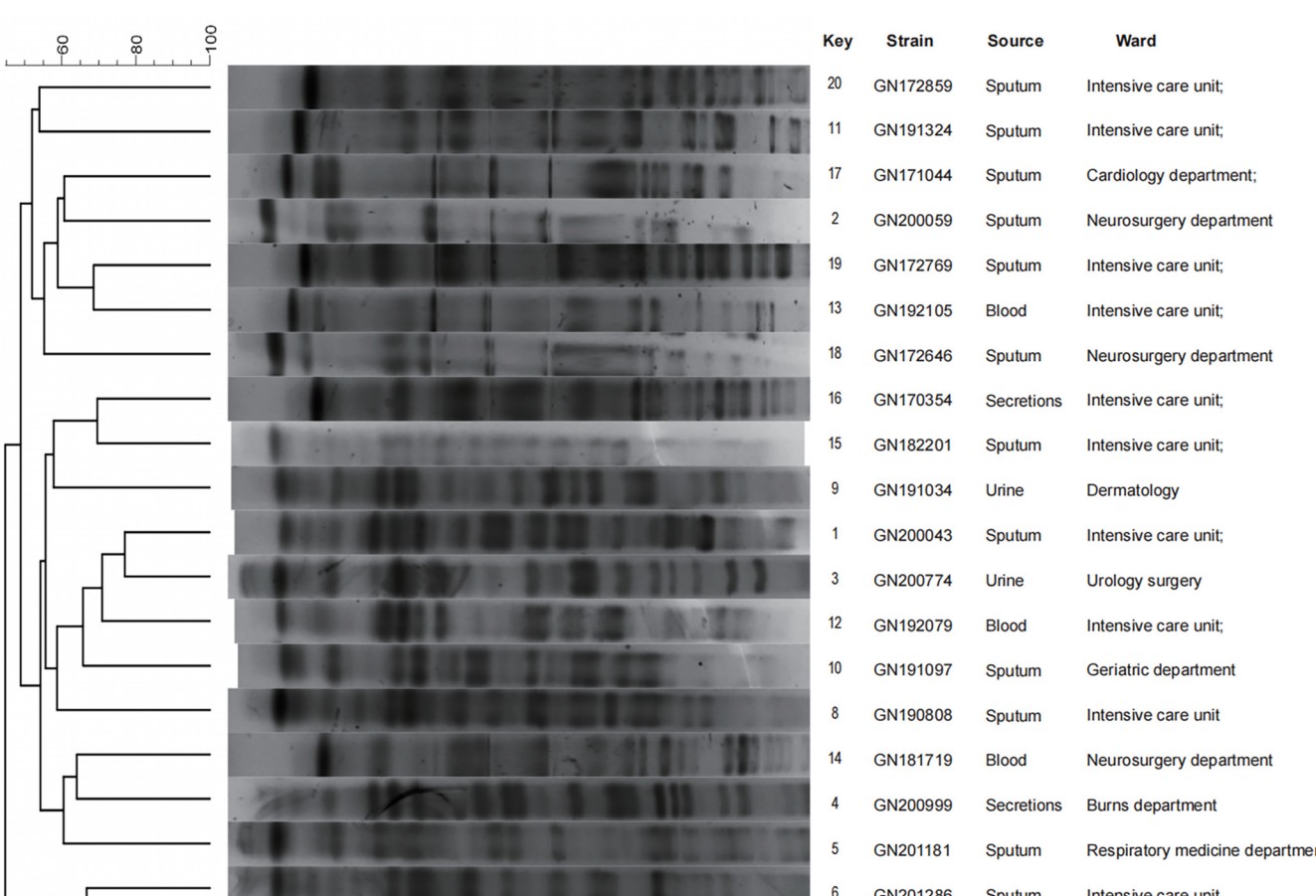

| Key | Strain | Source | Ward |
|---|---|---|---|
| 20 | GN172859 | Sputum | Intensive care unit; |
| 11 | GN191324 | Sputum | Intensive care unit; |
| 17 | GN171044 | Sputum | Cardiology department; |
| 2 | GN200059 | Sputum | Neurosurgery department |
| 19 | GN172769 | Sputum | Intensive care unit; |
| 13 | GN192105 | Blood | Intensive care unit; |
| 18 | GN172646 | Sputum | Neurosurgery department |
| 16 | GN170354 | Secretions | Intensive care unit; |
| 15 | GN182201 | Sputum | Intensive care unit; |
| 9 | GN191034 | Urine | Dermatology |
| 1 | GN200043 | Sputum | Intensive care unit; |
| 3 | GN200774 | Urine | Urology surgery |
| 12 | GN192079 | Blood | Intensive care unit; |
| 10 | GN191097 | Sputum | Geriatric department |
| 8 | GN190808 | Sputum | Intensive care unit |
| 14 | GN181719 | Blood | Neurosurgery department |
| 4 | GN200999 | Secretions | Burns department |
| 5 | GN201181 | Sputum | Respiratory medicine department |
| 6 | GN201286 | Sputum | Intensive care unit |
| 7 | GN202933 | Secretions | Burns department |

**FIG 1** Homology analysis. Pulsed-field gel electrophoresis (PFGE) cluster analysis of 20 clinical strains of multidrug-resistant *K. pneumoniae* from different sources. The dendrogram was developed using BioNumerics analysis software. Percent similarities are described by the unweighted pair group method using the arithmetic average (UPGMA) method. These experiments were performed at least three times, and representative results are shown.

pulsed-field gen electrophoresis (PFGE). Based on DNA fingerprinting patterns, the similarity coefficient of these strains was <80%, indicating that they were epidemiologically unrelated (Fig. 1).

**GaNt-colistin combined against MDR *K. pneumoniae*.** Previous reports have shown that gallium has significant *in vitro* antibacterial activity against *K. pneumoniae* clinical isolates (22) and can be used to synergize with colistin (polymyxin E) against *P. aeruginosa* (20). Therefore, we examined the combinatory bactericidal activity of GaNt-colistin against MDR *K. pneumoniae*.

To evaluate the role of colistin alone or combined with GaNt against the 20 selected MDR *K. pneumoniae* clinical isolates, we performed a spot dilution assay on M9CA plates supplemented with colistin, GaNt, or both. The sensitivity of CST-resistant *K. pneumoniae* to colistin was enhanced in the presence of GaNt, as evidenced by an increase in the number of bacteria required to obtain visible growth (Fig. 2A and B). Similar results were observed for CST-sensitive *K. pneumoniae* (Fig. S2A and B).

To confirm the effect of GaNt on the antimicrobial activity of colistin, we performed three independent assays. First, *K. pneumoniae* ATCC 43816, ATCC 13883, ATCC BAA-1705, ATCC 700603, and clinical strain GN 181608 (23) were chosen to perform a spot dilution assay. The combination of 0.25 $\mu$g/mL colistin and 10 $\mu$g/mL GaNt effectively curtailed *K. pneumoniae* growth (decreased by 3 to 4 $\log_{10}$ CFU/mL) (Fig. 3A). A similar phenomenon was observed when polymyxin B was used instead of colistin (Fig. S3) or

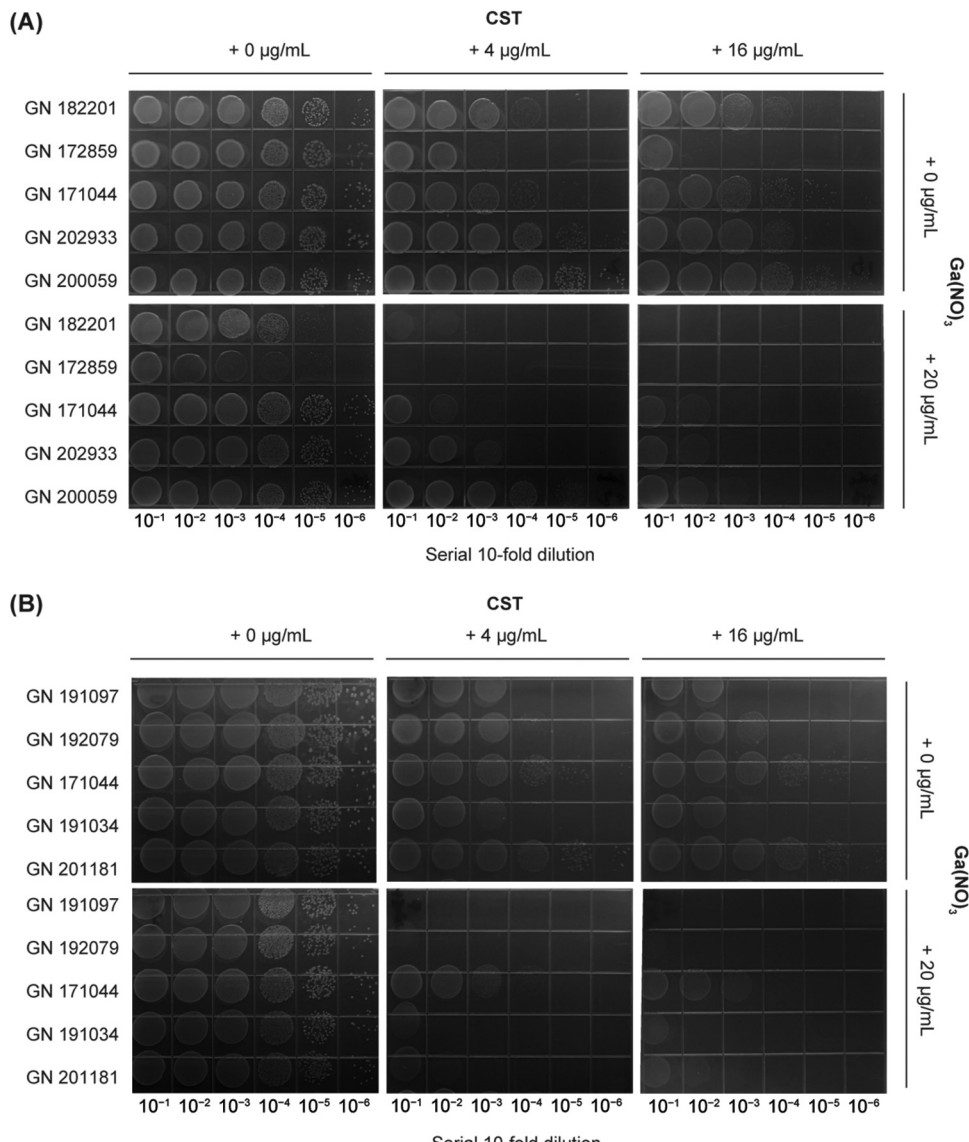

**FIG 2** *Klebsiella pneumoniae* sensitivity to colistin in the presence of GaNt. Growth of serial 10-fold dilutions of 10 colistin (CST)-resistant MDR *K. pneumoniae* clinical strains on solid medium containing 0 to 16 µg/mL colistin and 0 to 20 µg/mL GaNt. The plates were incubated at 37℃ for 20 h and photographed. (A and B) GN 182201, GN 172859, GN 171044, GN 202933, and GN 200059 were used in panel A, and GN 191097, GN 192079, GN 171044, GN 191034, and GN 201181 were used in panel B. These experiments were performed at least three times, and representative results are shown.

gallium maltolate replaced GaNt (Fig. S4). Together, these results indicated that GaNt enhanced the growth-inhibitory effect of colistin on *K. pneumoniae*.

K. pneumoniae ATCC 43816, as a well-studied strain capable of causing respiratory disease in mouse models (24–26), was chosen for further studies *in vivo* and *in vitro*. Consistent with the spot dilution assay (Fig. 3A), rapid killing (Fig. 3B) and bacterial growth curve assays (Fig. 3C) also showed that the addition of GaNt markedly decreased the growth of *K. pneumoniae* ATCC 43816. We also performed crystal violet staining of ATCC 43816 biofilms and assessed the relative biofilm biomass. The results showed that the combination of 2 µg/mL colistin and 20 µg/mL GaNt enhanced the clearance of *K. pneumoniae* preformed biofilms compared to colistin or GaNt alone (Fig. S5A and B).

**ROS involvement in GaNt-colistin-mediated bacterial killing.** Colistin can influence the redox balance in *K. pneumoniae*, and gallium can increase ROS accumulation in *E. coli* (27, 28). To explore the effect of GaNt on colistin-stimulated ROS accumulation,

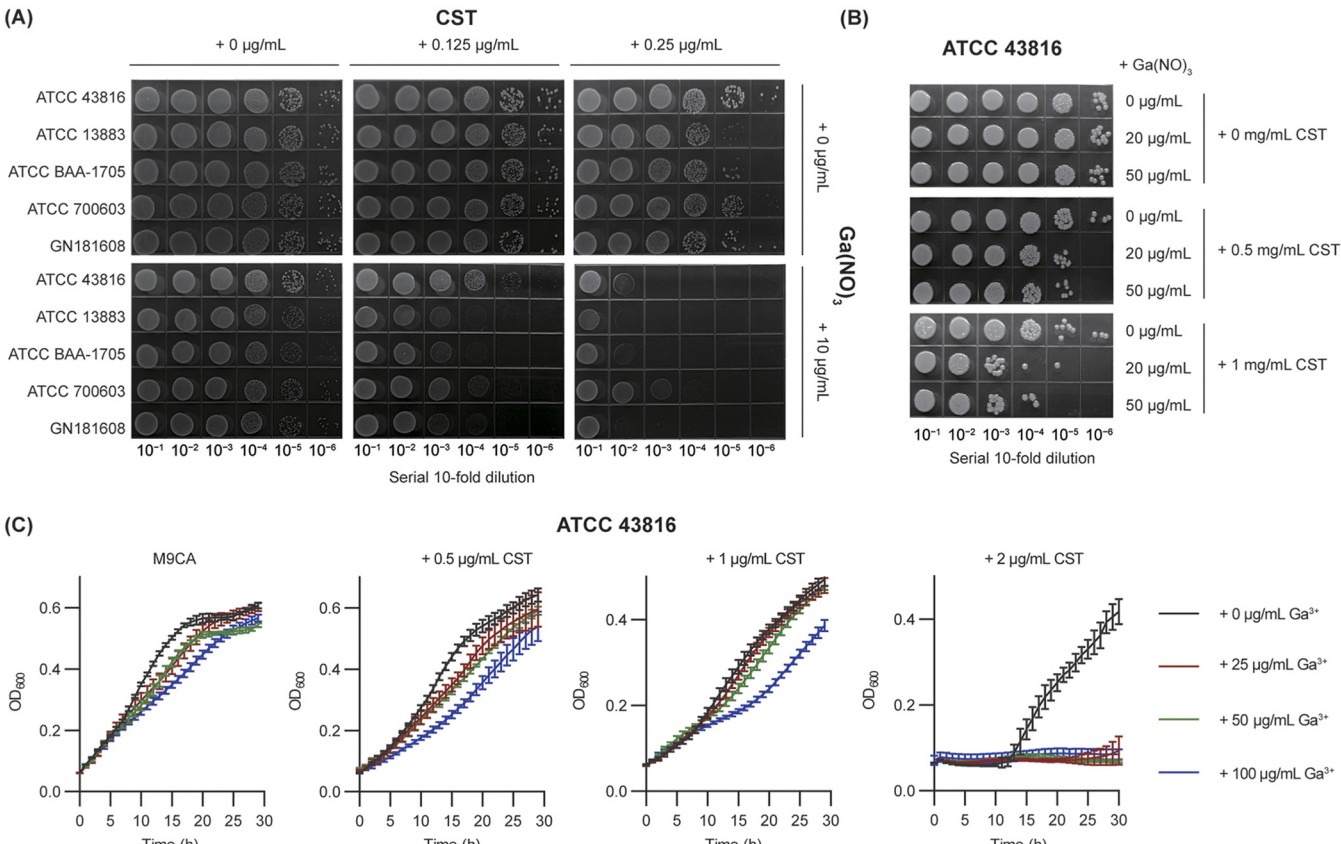

**FIG 3** The effect of colistin in combination with GaNt on *Klebsiella pneumoniae*. (A) Spot dilution assay. Serial 10-fold dilutions of cells of 4 wild-type standard and a clinical strain, GN 1816081, were spotted onto M9CA plates containing 0 to 0.25 $\mu$g/mL colistin (CST) and 0 to 10 $\mu$g/mL GaNt [Ga(NO)$_3$]. The plates were incubated at 37°C for 20 h and photographed. (B) Rapid killing assay. *K. pneumoniae* strain ATCC 43816 was cultured in M9CA medium supplemented with the indicated concentrations of GaNt (0 to 50 $\mu$g/mL) and colistin (0 to 1 $\mu$g/mL). After 2 h of culture, bacterial cell viability was determined by serial dilution and plating on solid medium. (A and B) The images are representative of three independent replicates. (C) Bacterial growth curve assay. *K. pneumoniae* strain ATCC 43816 grown in 96-well plates containing M9CA medium supplemented with the indicated concentrations of GaNt (0 to 50 $\mu$g/mL) and colistin (0 to 2 $\mu$g/mL). Data are presented as the mean $\pm$ standard deviation ($n$ = 4 biological replicates).

intracellular ROS levels were measured using carboxy-2′,7′-dichlorodihydrofluorescein diacetate (H2DCFDA) in *K. pneumoniae* ATCC 43816. Not surprisingly, there was a moderate increase in ROS production during GaNt or colistin treatment of *K. pneumoniae*, but comparative analysis showed that GaNt in combination with colistin significantly increased ROS accumulation (Fig. 4A). Moreover, the fluorescent reporter dye 3′-(p-hydroxyphenyl) fluorescein (HPF) was used to verify the combined effect of GaNt and colistin on hydroxyl radical formation. GaNt treatment induced a mild increase in ROS levels, similar to the effect of colistin treatment, while the GaNt-colistin combination caused a 3.9-fold increase in ROS accumulation (Fig. 4B).

Dimethyl sulfoxide (DMSO), a ROS scavenger, can protect bacteria from ROS-mediated bacterial death (29) in a rapid killing assay. Therefore, exogenous DMSO was added to examine the effect of GaNt on ROS-mediated colistin-induced programmed cell death. Coincubation with 1 or 2% DMSO sharply diminished this combination-mediated rapid bacterial killing (decreased by 4 log$_{10}$ CFU/mL killing) (Fig. 4E). Intracellular ROS levels were measured using carboxy-H2DCFDA and HFP, under the same experimental conditions. Both the assays showed that coincubation with DMSO reduced the intracellular ROS accumulation (Fig. 4C and D). Additionally, similar phenomena were observed when other exogenous ROS scavenging compounds (dimethyl thiourea and ascorbic acid) (30, 31) were added to the cultures (Fig. S6). Taken together, these observations suggest that enhanced ROS levels correlate with the increased antimicrobial activity of GaNt-colistin.

To further validate these results, we also performed rapid killing and ROS assays using another two *Klebsiella pneumoniae* strains (ATCC 700603 and GN 182201). Like

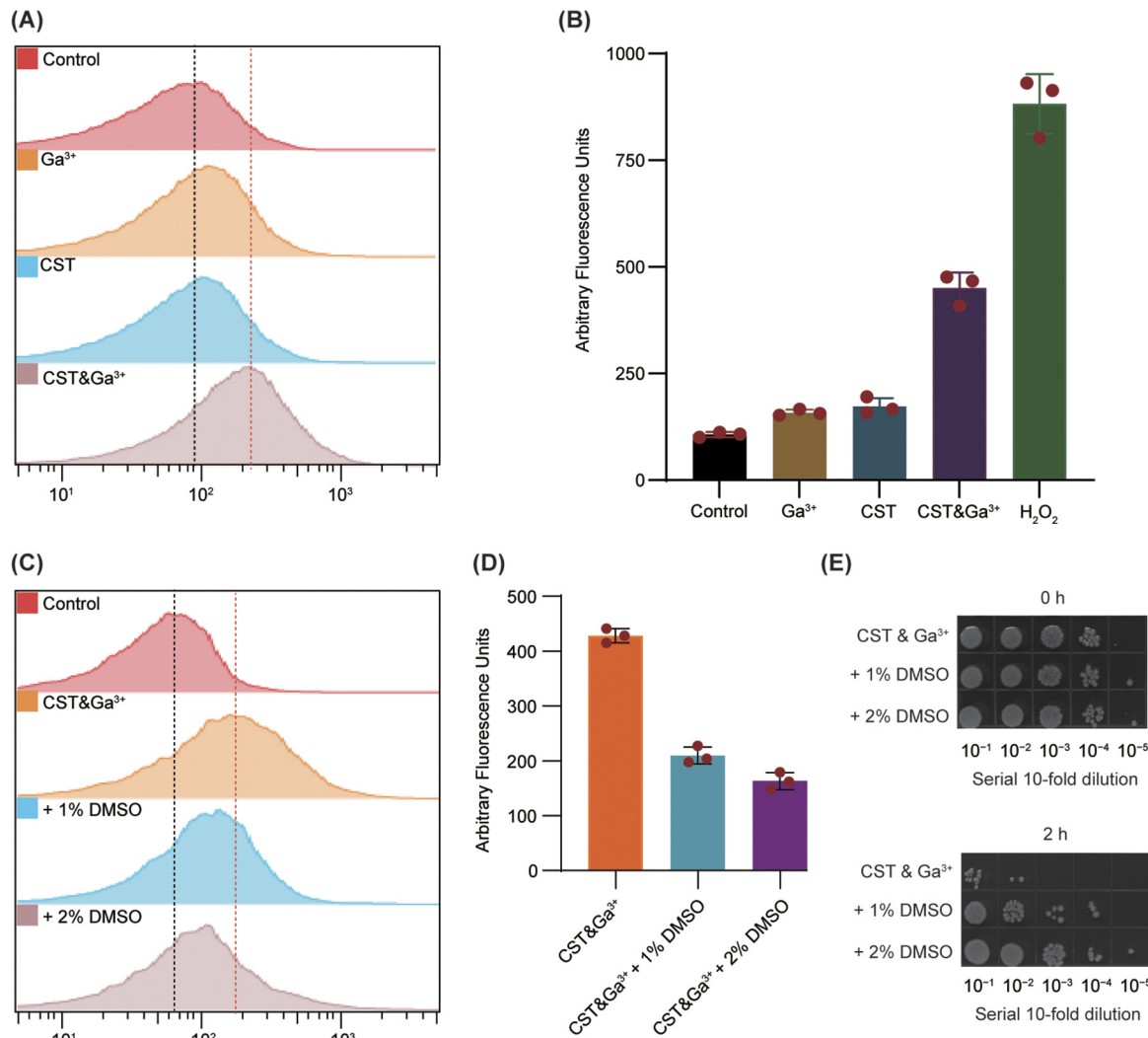

**FIG 4** GaNt treatment increased ROS accumulation, leading to bacterial cell death. (A) Wild-type *K. pneumoniae* ATCC 43816 was untreated (control) or treated with 2 $\mu$g/mL colistin (CST), 20 $\mu$g/mL GaNt (Ga$^{3+}$), 2 $\mu$g/mL colistin and 20 $\mu$g/mL GaNt (CST&Ga$^{3+}$), or 0.1% H$_2$O$_2$ (H$_2$O$_2$) for 2 h before samples were incubated with 10 mM carboxy-H2DCFDA for 30 min. Reactive oxygen species (ROS) were determined using flow cytometry. (B) Hydroxyl radical-specific fluorescent dye was added after treatment as in A, and the fluorescence was measured (490/515 nm). (C) Samples containing 2 $\mu$g/mL colistin and 20 $\mu$g/mL GaNt were supplemented without or with 1 or 2% DMSO, respectively, for 2 h before incubation. DMSO was used as a ROS scavenger. ROS were determined using flow cytometry. (D) Hydroxyl radical-specific fluorescent dye was added after treatment as in panel C, and the fluorescence was measured (490/515 nm). Data represent the means from three independent experiments, and error bars represent standard errors of the mean. (E) Serial 10-fold dilutions of ATCC 43816 cells were spotted onto M9CA plates for 0 and 2 h and were treated with 2 $\mu$g/mL colistin and 20 $\mu$g/mL GaNt combined or with 1 or 2% DMSO added, respectively. The plates were incubated at 37°C for 20 h and photographed. The images are representative of three independent replicate experiments.

ATCC 43816, ATCC 700603 and GN 182201 were protected by exogenous DMSO or ascorbic acid from GaNt-colistin-mediated bacterial killing (Fig. S7A and B) and intracellular ROS accumulation (Fig. S7C and D).

**GaNt suppresses bacterial antioxidant activity as revealed by RNA-seq.** Catalase and superoxide dismutase are key bacterial antioxidant enzymes that are widely utilized to modulate the response to oxidative stress (32). Because Ga ions can substitute for Fe in many biological systems, it has been thought to interfere with antioxidant enzymes (e.g., catalase and superoxide dismutase) and induce ROS overproduction in bacteria (18). Because the increased ROS accumulation resulting from GaNt treatment is important for the antimicrobial activity of colistin, we sought to study the relationship between ROS accumulation and GaNt treatment. To gain a deeper understanding of the molecular mechanisms of GaNt and induced gene expression changes at the

mRNA level, we performed a transcription analysis of *K. pneumoniae* ATCC 43816 after exposure to GaNt or the GaNt-colistin combination for 2 h.

In this study, the expression profiles of differentially expressed genes (DEGs) were expected to meet the following two criteria: (i) the value of the $\log_2$ fold change ($\log_2$FC) was $\geq$1, and (ii) the adjusted $P$ value was <0.05. As shown in Table S3, we found that 432 genes were induced by GaNt treatment and 631 genes were repressed. For global functional analysis of DEGs, gene ontology (GO) annotation was performed using Blast2GO. All corresponding proteins were associated with at least one GO term and grouped into 27 groups based on GO level 2 classification. Consistent with previous observations in other bacteria, GaNt treatment (in the absence of Fe limitation) induced some changes in the expression of Fe-responsive genes in *K. pneumoniae*. More importantly, GaNt treatment repressed the expression of genes involved in the antioxidant activity of *K. pneumoniae* (Fig. 5A). In particular, the transcriptional levels of two superoxide dismutase-encoding genes, *sodB* and *sodC*, and two catalase-encoding genes, *katE* and *katG*, were downregulated (Fig. 5C). Next, the comparison of treatment with colistin in combination or alone revealed an upregulation of 574 and downregulation of 752 DEGs, respectively (Table S4). Among the downregulated genes, seven were involved in bacterial antioxidant activity; *sodB* (10.57-fold), *sodC* (3.59-fold), *katE* (2.43-fold), and *katG* (21.39-fold) were also observed (Fig. 5B and D). A similar result was obtained when the expression of genes (*sodABC* and *katEG*) encoding antioxidant enzymes was examined by quantitative reverse transcription-PCR (qRT-PCR) (Fig. S8). These observations support the notion that genes involved in antioxidant activity of *K. pneumoniae* could be suppressed by GaNt with or without colistin, which appears to cause intracellular ROS accumulation.

**GaNt enhanced the growth-inhibitory effect of oxidative stress on *K. pneumoniae*.** To further examine whether the growth-inhibitory effect of ROS accumulation on *K. pneumoniae* is affected by GaNt, five *K. pneumoniae* strains in Fig. 3 were chosen to perform a spot dilution assay in which GaNt was included in the solid medium and hydrogen peroxide ($H_2O_2$) was added to the agar as a source of oxidative stress. In this assay, all *K. pneumoniae* strains were exposed to subinhibitory concentrations of gallium, and their sensitivity to $H_2O_2$ at a concentration range of 0 to 100 $\mu$M was measured. The result shows that adding GaNt (10 and 30 $\mu$g/mL) significantly increased the susceptibility of *K. pneumoniae* to $H_2O_2$ (Fig. 6). As $H_2O_2$ can induce ROS generation and cause extensive disruption of bacterial cells, we concluded that the presence of GaNt increased the sensitivity of *K. pneumoniae* to oxidative damage caused by $H_2O_2$.

Given that gallium influences other bacterial growth, we next tested the effect of the GaNt-$H_2O_2$ combination against *A. baumannii* (Fig. S9A), *E. coli* (Fig. S9B), and *P. aeruginosa* (Fig. S9C) in M9CA media supplemented with or without GaNt and $H_2O_2$. Consistent with *K. pneumoniae* strains, under subinhibitory concentrations of GaNt (20 $\mu$g/mL used in *A. baumannii*, 20 $\mu$g/mL used in *E. coli*, and 2.5 $\mu$g/mL used in *P. aeruginosa*), the growth of the indicated bacteria was inhibited by 50 $\mu$M $H_2O_2$, which was enhanced by GaNt. We suspect that this phenotype resulted from GaNt influencing the sensitivity of the bacteria to oxidative stress caused by $H_2O_2$.

**Iron-limiting conditions enhance the combined activity of GaNt-colistin.** The fact that host defenses severely limit availability indicates that invading organisms are often faced with extreme iron limitation during growth *in vivo* (33). To mimic the iron-poor environment for GaNt activity testing, the addition of 75 $\mu$M 2,2'-dipyridyl (DIP) to M9CA medium ensured an iron-deficient environment. Growth was then evaluated for the entire collection of the four ATCC strains in M9-DIP using a microtiter plate assay. All strains showed sufficient growth in M9-DIP medium. As expected, the addition of 2.5 $\mu$g/mL GaNt to M9-DIP increased the sensitivity of *K. pneumoniae* to colistin and $H_2O_2$. The concentrations of GaNt and colistin in M9-DIP were lower than those in M9CA (Fig. 7). A similar result was observed in the 20 *K. pneumoniae* clinical isolates (Fig. S10 and S11). Based on these results, it was concluded that iron deficiency significantly increased the combined antimicrobial activity of GaNt-colistin.

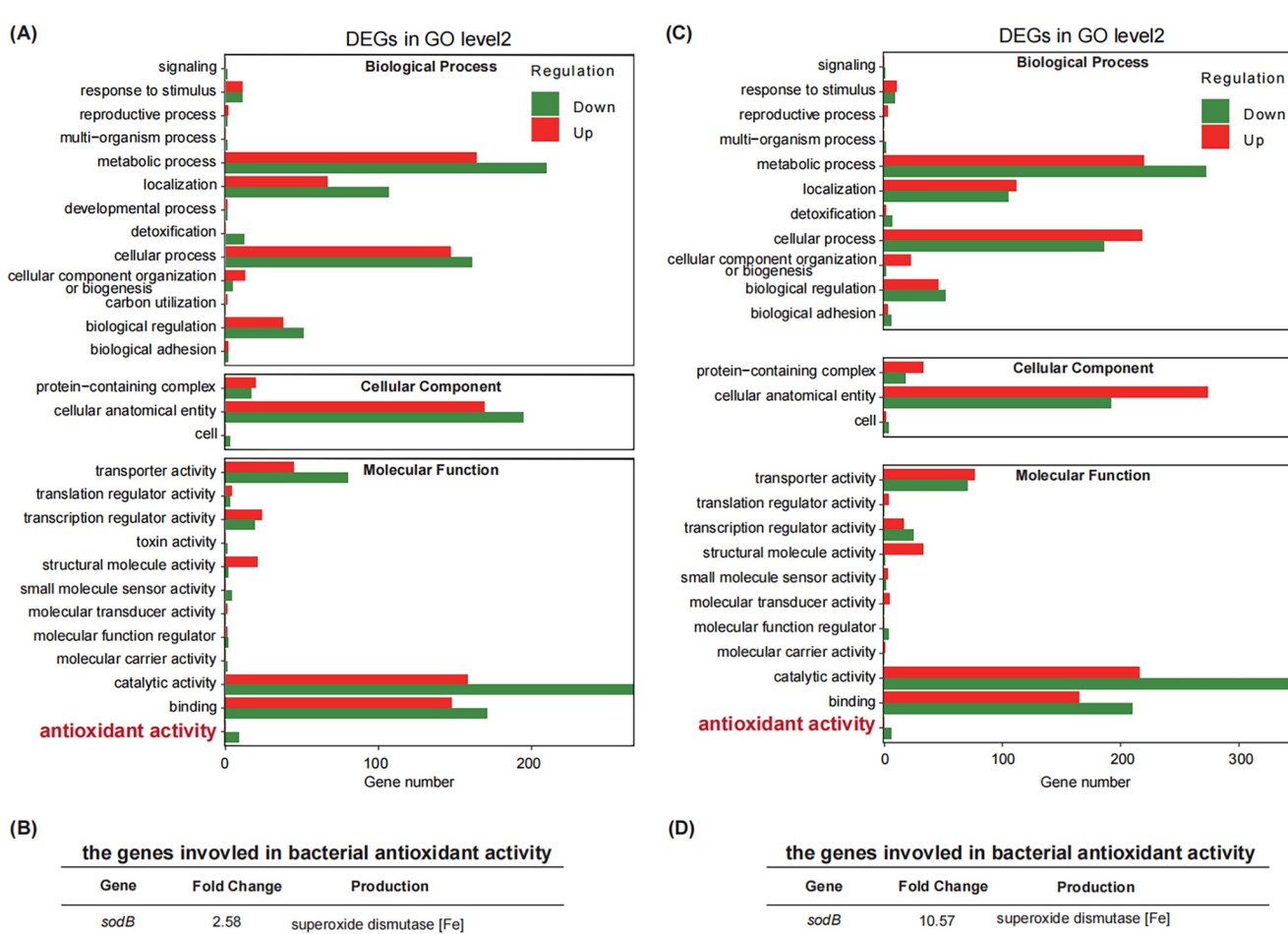

**FIG 5** The effect of GaNt on bacterial antioxidant activity as revealed by RNA-seq. (A and B) To identify global transcriptional changes, samples were untreated or treated with 5 μg/mL GaNt (A) or treated with 0.5 μg/mL colistin supplemented without or with 5 μg/mL GaNt (B) for 2 h before incubation. Level 2 GO annotation of upregulated and downregulated genes was done. We divided the sets into three major GO ontologies: biological process, cellular component, and molecular function. Red and green bars represent upregulated and downregulated genes, respectively. (C and D) The differentially expressed genes involved in bacterial antioxidant activity panel A are shown in panel C and those in panel B are shown in panel D.

**GaNt enhances the antimicrobial efficacy of colistin in murine infection models.**

To assess the antibacterial effects of GaNt and colistin *in vivo*, a mouse model of *K. pneumoniae*-induced lung injury was established (Fig. 8A). Treatment with colistin alone reduced the cell count by ~10-fold compared to that of the control group, while a significant difference was noted in the pulmonary bacterial load between the GaNt-colistin and colistin groups (Fig. 8B). We further quantified lung pathological sections and semiquantitative scores after *K. pneumoniae* infection. All the infected mice showed histological evidence of severe pneumonia, as demonstrated by enhanced alveolar wall destruction, interstitial inflammation, endothelialitis, and edema. Compared with the corresponding monotherapy group, the combination group had fewer inflammatory cells and less alveolar wall destruction (Fig. 8C and D). These results indicate that *K. pneumoniae*-induced lung inflammation can be alleviated by GaNt-colistin treatment.

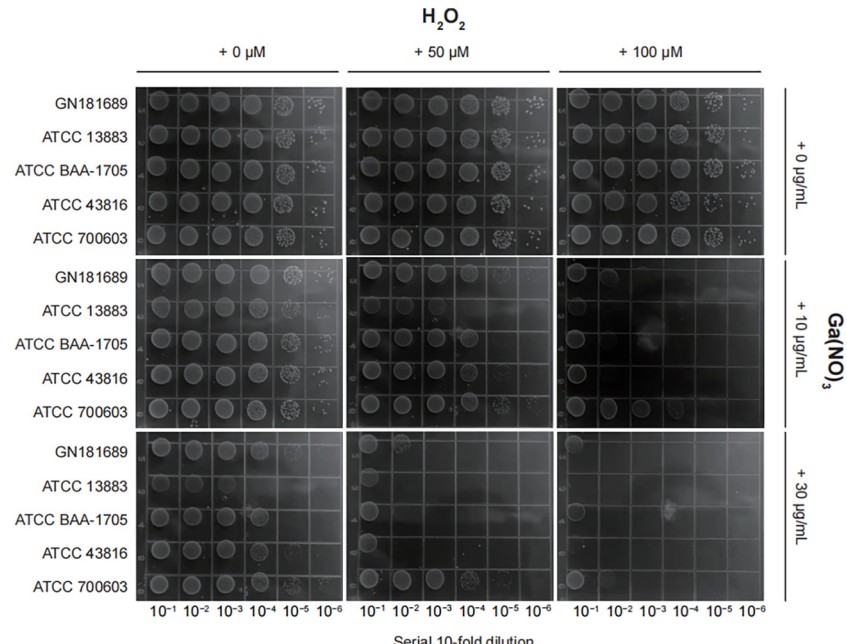

**FIG 6** GaNt increased the susceptibility of *Klebsiella pneumoniae* to $H_2O_2$. Serial 10-fold dilutions of cells of 4 wild-type standard strains and a clinical strain, GN 1816081, were spotted onto M9CA plates containing 0 to 30 $\mu$g/mL GaNt and 0 to 100 $\mu$M $H_2O_2$. The plates were incubated at 37°C for 20 h and photographed. The images are representative of three independent replicate experiments.

## DISCUSSION

The increase in multidrug-resistant *Enterobacteriaceae* has led to the renewed use of colistin, but the expression of mobile colistin resistance (MCR) enzymes (e.g., EptA, MCR-1, MCR-2, etc.) (34) or nonmobile colistin resistance (NMCR) enzymes (e.g., NMCR-1, NMCR-1.2, NMCR-1.8, etc.) (35) is shown to remodel the bacterial surface, which has caused the emergence of colistin resistance among Gram-negative bacteria worldwide (36). Despite these enormous risks, the discovery and development of antibiotics are difficult. GaNt has been approved for treating hypercalcemia associated with malignancy and has been explored for repurposing as an antibacterial agent (37, 38). However, knowledge of GaNt antimicrobial activity against *K. pneumoniae* is limited. A potential method to improve the GaNt effect is to combine it with a conventional antimicrobial agent. The identification of additional antibiotics that exhibit synergistic or additive interactions with GaNt is crucial for

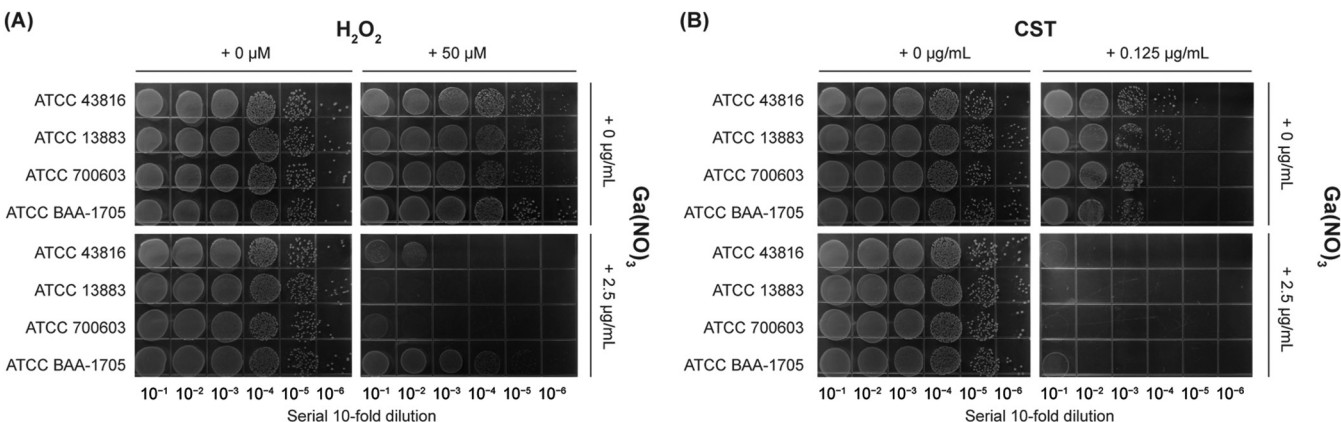

**FIG 7** Under iron-limiting conditions, GaNt increases the sensitivity of *Klebsiella pneumoniae* to colistin and $H_2O_2$. (A) Serial 10-fold dilutions of cells of 4 wild-type standard *K. pneumoniae* strains were spotted onto M9CA with 75 $\mu$M DIP plates containing 0 to 50 $\mu$M $H_2O_2$ and 0 to 2.5 $\mu$g/mL GaNt. (B) The samples in panel A were also spotted onto M9-DIP plates containing 0 to 0.125 $\mu$g/mL colistin and 0 to 2.5 $\mu$g/mL GaNt. All plates were incubated at 37°C for 20 h and photographed. These experiments were performed on at least three independent occasions, and representative results are shown.

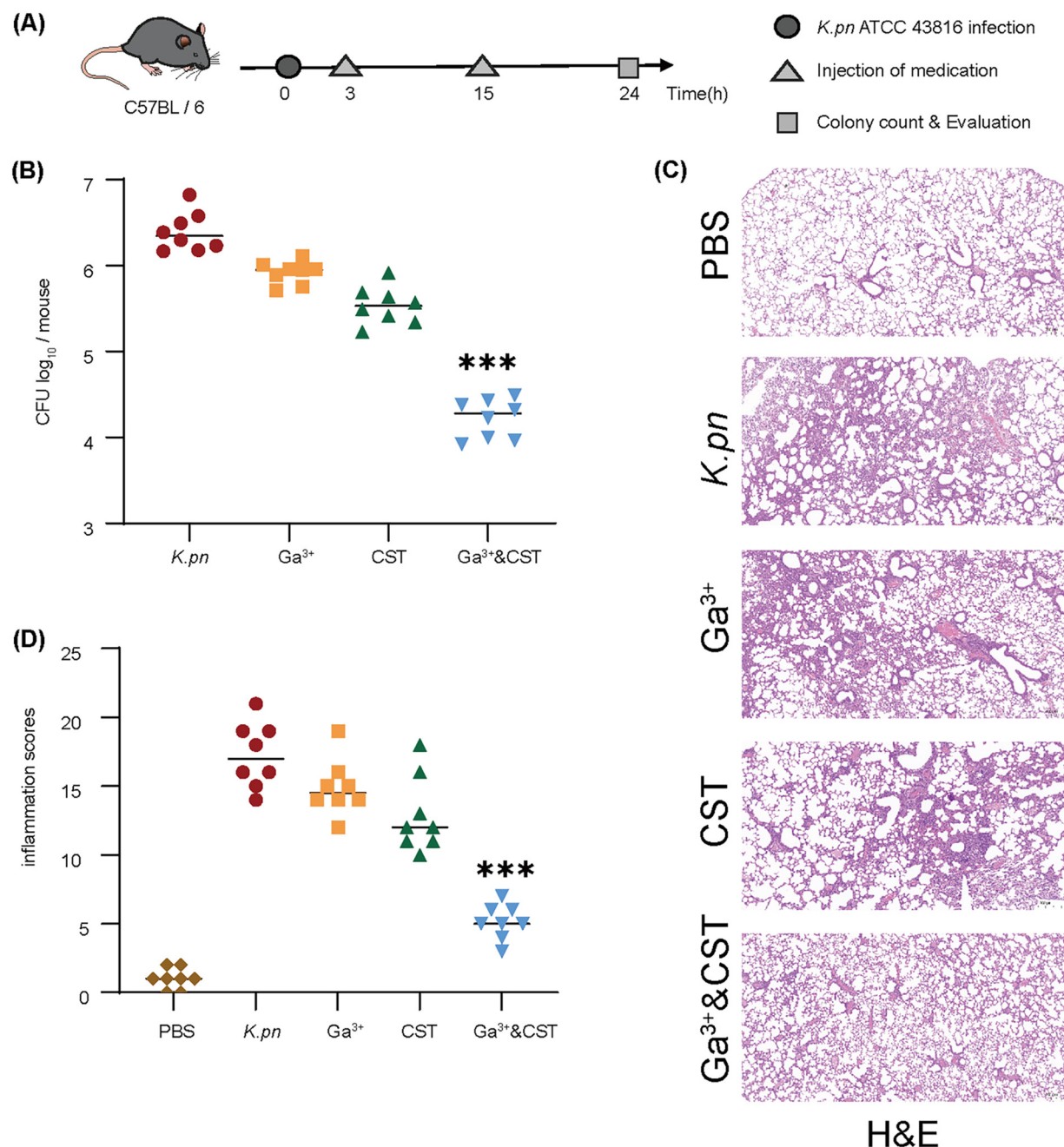

**FIG 8** Effect of colistin and GaNt on *Klebsiella pneumoniae* ATCC 43816-induced lung injury. (A) Schematic diagram of lung infection and treatment process. C57BL/6 mice were infected with $10^6$ CFU of *K. pneumoniae* via the intranasal route. Then, 3 h after infection, the mice were treated with 15 mg/kg GaNt ($Ga^{3+}$ group), 5 mg/kg colistin (CST group), or a combination of GaNt and colistin ($Ga^{3+}$&CST group) by intraperitoneal injection. The PBS group was used as a control. Another group (no infection group) that was injected with 50 $\mu$L of sterile PBS instead of *K. pneumoniae* was used as a negative control. Treatment was administered twice. The group size was 8 mice. (B) After 24 h of infection with *K. pneumoniae*, the pulmonary bacterial load was measured in the antibiotic-treated and -untreated control groups. The CFU of *K. pneumoniae* was counted in the infected mouse lung tissues after treatment with PBS, GaNt (15 mg/kg), colistin (5 mg/kg), or their combination 24 h postinfection. ***, $P < 0.001$ versus the *K. pneumoniae*-infected control. Data are from three independent experiments. (C) Representative lung slides of untreated and antibiotic-treated cells after *K. pneumoniae* infection; total lung histopathology scores. ***, $P < 0.001$ versus the *K. pneumoniae*-infected control. H&E staining; Scale bars = 200 $\mu$m (4×). The image is representative of three independent experiments.

clinical applicability (20, 39). In this study, we established that GaNt can suppress bacterial antioxidant activity and induce ROS accumulation, thereby improving the antimicrobial activity of colistin against *K. pneumoniae*.

The potential of combining GaNt with antibiotics against the opportunistic human pathogen *P. aeruginosa* was explored (20, 40). The results showed that the addition of

GaNt to antibiotics restored growth inhibition in most antibiotic-resistant clones, indicating that GaNt has the potential to be used as an antibiotic adjuvant. Interestingly, we found that in the presence of GaNt, the growth of MDR *K. pneumoniae* clinical isolates was significantly inhibited by colistin. To evaluate the effect of GaNt on the antimicrobial activity of colistin, three independent assays were performed to measure the combined activity of gallium and colistin: spot dilution and rapid antimicrobial time-kill assays and growth curve inhibition tests. The results showed that even when GaNt did not directly contribute to growth inhibition, the susceptibility of *K. pneumoniae* to colistin *in vitro* was significantly enhanced. Thus, further research is required to identify the effect of GaNt on colistin activity against *K. pneumoniae*.

A recent study suggested that a GaNt-mediated surge of intracellular ROS is involved in killing *E. coli* (28), which led us to hypothesize that GaNt can also induce ROS accumulation in *K. pneumoniae*. However, we found that GaNt alone had a weak effect on intracellular ROS levels, similar to that of colistin treatment, but significantly increased ROS accumulation in combination with colistin. Although the contribution of ROS to bacterial killing remains controversial (41, 42), the results from this study showed that DMSO is a ROS scavenger, reducing the combined antimicrobial activity of GaNt-colistin, supporting an important role for ROS in GaNt-colistin-mediated killing. Similarly, after the addition of DMSO, combination treatment was still more effective than colistin monotherapy, indicating that mechanisms aside from ROS regulation may be involved.

To gain greater insight into this process, we performed RNA-seq analysis of *K. pneumoniae* ATCC 43816 cultured on M9CA medium in the presence and absence of a subinhibitory concentration of GaNt (0.5 $\mu$g/mL) to examine the effect of GaNt on the gene expression profile. In *P. aeruginosa*, RNA-seq analysis revealed that GaNt repressed genes involved in the oxidative stress response, including *sodM* and *PA4469* (33). In *K. pneumoniae*, genes involved in bacterial antioxidant activity, including *sodB*, *sodC*, *katE*, *katG*, *pcaC*, *IT767_RS09590*, *IT767_RS13980*, *IT767_RS02465*, and *IT767_RS06360* were downregulated by GaNt. These data suggest that GaNt is an oxidative stress-inducing compound. When GaNt was added to M9CA medium containing colistin, *sodB*, *sodC*, *katE*, and *katG* were strongly repressed, whereas *gorA* and *IT767_RS00785* were moderately repressed. A possible explanation for the change in gene expression is that polymyxins caused cell envelope remodeling and induced oxidative stress in bacteria, which affected the ability of GaNt to induce oxidative stress in *K. pneumoniae*. Consistent with this hypothesis, we observed that colistin enhanced the antibiofilm activity of GaNt. Further studies are required to reveal the detailed mechanism underlying the effect of colistin on GaNt activity.

GaNt-mediated inhibition of bacterial antioxidant activity makes *K. pneumoniae* more susceptible to killing by $H_2O_2$, similar to *A. baumannii*, *E. coli*, and *P. aeruginosa* (8, 9, 28). Since invading bacteria are exposed to extreme iron limitation *in vivo*, it was also relevant to examine the antibacterial activity of GaNt-colistin in iron-depleted cultures (43). To this end, the addition of DIP strongly inhibited bacterial growth in M9CA medium, suggesting that iron-limiting conditions in bacterial cultures enhance the combined antimicrobial activity of GaNt-colistin. GaNt-colistin also caused a dramatic reduction in *K. pneumoniae* lethality in murine lung infections. Remarkably, these findings highlight that the GaNt-colistin combination could serve as a potential therapeutic strategy for lung infection.

**Conclusions.** In summary, we demonstrated that GaNt could induce excessive accumulation of intracellular ROS and potentiate the antibacterial activity of colistin against bacteria *in vitro* and *in vivo*. Nevertheless, further studies are required to elucidate the mechanism of action of GaNt, which should help us to develop novel GaNt-based strategies for the treatment of infections caused by drug-resistant *K. pneumoniae* strains.

## MATERIALS AND METHODS

**Bacterial isolates, antimicrobials, and mice.** From January 2017 to December 2020, 1,200 nonrepetitive *K. pneumoniae* strains were isolated from clinical samples of inpatients and outpatients from the Anhui Center for Surveillance of Bacterial Resistance with qualified data (44). These strains were identified using a matrix-assisted laser desorption–ionization time of flight mass spectrometry (MALDI-TOF MS) automated microbiology system (Bioyong, China) (45). All isolates were stored in Muller-Hinton broth (MHB; Sigma-Aldrich, USA) with 50% glycerol in cryovials for stock at −80°C and grown on Muller-

Hinton agar (MHA; Sigma-Aldrich) at 37°C. *K. pneumoniae* ATCC 43816, ATCC 13883, ATCC 700603, and ATCC BAA-1705, *A. baumannii* ATCC 17978 and ATCC 19606, *E. coli* ATCC 25922 and BW 25113, and *P. aeruginosa* MPAO1 and CMCC 10104 were the wild-type standard strains.

All the antibiotics, Ga(NO$_3$)$_3 \cdot$ xH$_2$O, gallium maltolate, dimethyl thiourea, and ascorbic acid, were obtained from Sigma-Aldrich (St. Louis, MO, USA). All prepared solutions were stored at −20°C for up to 1 month.

Wild-type female C57BL/6 mice (8 weeks, 16 to 20 g) were purchased from the Experimental Animal Center of the Anhui Province (Hefei, China). All experiments involving mice were approved by the Institutional Animal Care and Use Committee of Anhui Medical University (approval no. LLSC20190253).

**Susceptibility testing by Agar dilution method.** A total of 24 different antibiotics (see details in Table S1) were used in the agar dilution method (46). First, 1 mL of the serial dilutions of extracted antibiotics was added to 19 mL of still-liquid MHA medium containing <2.2 mg/L iron (47) at a temperature of 45 to 50°C under aseptic conditions to perform growth tests of the analyzed strains. The supplemented media were homogenized and poured on petri dishes with a diameter of 9 cm. Then a 0.5 McFarland (McF) suspension was diluted 1:10 in 1× phosphate-buffered saline (PBS), 1 $\mu$L of which was inoculated on the prepared plates using an HMI-60&24 multipoint inoculator (HengAo Technology Ltd., Tianjin, China), resulting in a final bacterial inoculum of $1 \times 10^4$ CFU/spot. To ensure the reliability of the experiment, each plate was set with a blank and a standard *E. coli* ATCC 25922 strain control. After 30 min of inoculation, plates were incubated at 36°C for 18 to 20 h. MICs were defined as the lowest drug concentrations that inhibited visible bacterial growth. Susceptibility was determined using breakpoints from the Clinical and Laboratory Standards Institute (CLSI) (48).

**Pulsed-field gel electrophoresis (PFGE).** A total of 10 CST-resistant (MIC ≥ 8 $\mu$g/mL) and 10 CST-sensitive strains (MIC ≤ 1 $\mu$g/mL) (see details in Table S1) were selected from the *K. pneumoniae* strains described above and subjected to PFGE. For the PFGE electrophoresis, the restriction endonuclease utilized was XbaI (TaKaRa, lot no. AIF2232A), and the DNA size marker used was *Salmonella* H9812. All strains were prepared by gelatinization, enzymatic digestion, and electrophoresis. Then, the PFGE images were processed by BioNumerics software, and the tree diagram was drawn. The similarity coefficient of the strains in the similarity analysis matrix (>80%) were of the same PFGE type (49).

**Spot dilution assay.** A spot dilution test was carried out as described previously (50) with few adjustments. M9CA medium (Sangon Biotech, China) containing 0.07 mg/L iron determined by inductively coupled plasma optical emission spectrometry (ICP–OES) at Shiyanjia lab (www.Shiyanjia.com) was used as a growth medium for the cultivation of bacteria in this assay. Briefly, bacteria were cultured in M9CA medium with shaking (250 rpm) at 37°C overnight. Serial 10-fold dilutions in 1× PBS were prepared, and 10 $\mu$L of each dilution was spotted onto M9CA plates containing the test compound at the specified concentrations. The plates were imaged after 18 to 20 h of incubation at 37°C.

**Growth curve determination.** Briefly, overnight M9CA cultures were diluted to an optical density at 600 nm (OD$_{600}$) of 0.05 in M9CA medium; a 100-$\mu$L volume of the sample supplemented without or with GaNt, colistin, or both was added to the wells, and subsequently, a 50-$\mu$L volume of filter-sterilized mineral oil was added to prevent evaporation during the assay. The cultures were grown at 37°C in an automatic microplate reader (Tecan, Switzerland). OD$_{600}$ readings were measured every 60 min.

**Rapid killing assay.** Rapid antimicrobial killing assays were conducted using colistin alone or with GaNt according to a previously described method (51). ATCC 43816 was cultivated for 14 to 16 h in 5 mL M9CA. The cultures were diluted to 1:100 with 5 mL of fresh broth and grown to an OD$_{600}$ of 0.25 to 0.3 (log-phase) on a shaker at 37°C. ATCC 43816 was mono-treated with 0.5, 1, and 2 mg/L colistin, 20 and 50 mg/L GaNt, or a combination of colistin and GaNt, and bacteria treated with DMSO were also evaluated. Samples were collected at 0 and 2 h and then diluted 10- to 10$^6$-fold with 1× PBS. The diluted samples were aseptically placed (10 $\mu$L) on agar plates incubated at 37°C for 16 h to measure the viable CFU/mL. Controls (untreated) were used in each experiment. Each experiment was performed in triplicate.

**Determination of biofilm biomass by crystal violet assay.** An ATCC 43816 suspension (0.1 mL, 10$^6$ CFU/mL) was placed in the wells of a 96-well plate and cultured in a 37°C incubator for 24 h. After the bacterial biofilms were formed, they were treated with 2 $\mu$g/mL colistin, 20 $\mu$g/mL GaNt, or both. The incubation was then continued at 37°C for 12 h to allow further bacterial biofilm growth or ablation. Removing the medium and suspended bacteria, the wells were rinsed with 1× PBS without destroying the established biofilm. After being fixed with methanol for 10 min, the bacterial biofilms were stained with 0.1% crystal violet solution. The wells were cleaned with 1× PBS to remove extra color 10 minutes later. After drying, the crystal violet-stained biofilms were assessed upon addition of 33% acetic acid and OD$_{590}$ measurement.

**ROS and hydroxyl radical assay.** To assess intracellular ROS accumulation, the fluorescent probe carboxy-H2DCFDA (Thermo Fisher, Waltham, MA, USA) was utilized. Since peroxynitrite was absent in the medium, as *K. pneumoniae* was not exposed to acidified nitrite, the possibility of the formation of hydroxyl radical from peroxynitrite was ruled out (52). 3'-(*p*-Hydroxyphenyl) fluorescein (HPF) was used for the detection of hydroxy radical, although it is oxidized by hydroxyl radical and peroxynitrite (53). Exponentially growing cultures (ATCC 43816) were treated with colistin (2 $\mu$g/mL) alone, GaNt (20 $\mu$g/mL) alone, or combination for 2 h. The treated and untreated samples were then incubated with carboxy-H2DCFDA or HPF at a final concentration of 10 $\mu$M for 30 min for detection of intracellular ROS. Samples (200 $\mu$L) were washed twice with precooled 1× PBS to scavenge the agents. Flow cytometry (BD FACSCelesta; Franklin Lakes, NJ, USA) was used to examine 100,000 cells. The intensity of bacterial fluorescence was measured using a multimode microplate reader (490/515 nm) (54). As a positive control, cultures were treated with hydrogen peroxide (H$_2$O$_2$), which was previously demonstrated to induce hydroxyl radicals (55).

To further test DMSO as a protection against ROS (peroxide)-mediated rapid killing in growth kinetics, cultures were treated with 2 $\mu$g/mL colistin and 20 $\mu$g/mL GaNt supplemented without or with 1 or 2% DMSO. The same method was used for detection. Data were analyzed using FlowJo software version 10.4 (Ashland, OR, USA).

**RNA isolation, RNA-seq, and RNA-seq data analysis.** ATCC 43816 was cultured for 14 h and diluted 50-fold in M9CA medium with a subinhibitory concentration of 5 $\mu$g/mL GaNt and 0.5 $\mu$g/mL colistin or both. Liquid cultures (20 mL) were cultivated in a 100-mL flask at 37°C for 2 h with shaking at 250 rpm. RNA extraction, rRNA removal, cDNA library construction, and paired-end sequencing with the Illumina HiSeq 2000 system were completed by Guangdong Magigene Biotechnology Co. Ltd. DEGs were detected using the edgeR software package. A fold change of ≥2 and a false-discovery rate (FDR) of ≤0.05 (edgeR, Benjamini-Hochberg method) were used as thresholds to determine the DEGs.

**Quantitative real-time PCR (qRT-PCR) analysis.** Bacterial gene expression analysis was performed by qRT-PCR as described previously (56). The total DNase-treated RNA was extracted using an RNeasy kit (Qiagen, Hilden, Germany) and was reverse-transcribed to synthesize cDNA using a PrimeScript cDNA synthesis kit (TaKaRa, Kyoto, Japan) according to the manufacturer's recommendations. qRT-PCR of *sodA*, *sodB*, *sodC*, *katE*, and *katG* was performed using PrimeScript RT master mix (TaKaRa) with a three-step real-time PCR system (LightCycler 96; Roche, Basel, Switzerland). The amplicon of *rrsE6* was used as an internal control. The primers for qRT-PCR are listed in Table S2. The expression levels of the target genes of interest were determined by the $2^{-\Delta\Delta CT}$ calculation method and reported as fold change values. Reactions were run in triplicate.

**Murine lung infection model.** Based on previous studies (57), we generated lung infection models to evaluate the effect of GaNt on antibacterial activity *in vivo*. C57BL/6 mice (8 per group) were anesthetized with isoflurane and inoculated intranasally with $1\times 10^6$ CFU of *K. pneumoniae* ATCC 43816 in 50 $\mu$L of sterile PBS. After 3 h and 15 h, they were treated with colistin alone (intraperitoneally [i.p.] at 30 mg/kg/day), GaNt alone (i.p. at 3 mg/kg/day), or a combination of colistin and GaNt. Two control groups, blank control (no treatment) group and solvent control group (PBS), were set for each experiment.

Mice were euthanized by $CO_2$ asphyxiation at 24 h postinfection, and lung tissue was removed for later analysis. To determine bacterial density, lungs were homogenized in 1 mL $1\times$ PBS and transferred into a sterile test tube for serial dilution; 100 $\mu$L was cultured onto broth agar for viable counting, which was performed the next day following overnight incubation at 37°C. For histological examination, paraformaldehyde-fixing solution was infused into the lungs, and then 4-$\mu$m sections of embedded tissues were deparaffinized and stained with hematoxylin and eosin (H&E). Pathology scoring was based on a previously described method (58). Briefly, the total lung inflammation score was counted as the sum of the scores for each parameter, the maximum being 24.

**Statistical analysis.** The data represent the means from three independent experiments, and error bars represent standard errors of the means. Comparisons between groups were performed using Student's *t* test and the Kruskal-Wallis one-way analysis of variance (ANOVA) test. Statistically significant differences are indicated as follows: *, $P < 0.05$; **, $P < 0.01$; ***, $P < 0.001$; ****, $P < 0.0001$. All graphs were generated using Prism version 8.0 (GraphPad, Inc., San Diego, CA, USA), FlowJo version 10.4 (Ashland, OR, USA), and the Illustrator CC 2021 (Adobe Systems, Inc., USA).

**Ethics approval.** The animal study was reviewed and approved by the Animal Experimentation Ethics Committee of Anhui Medical University (approval no. LLSC20190253), and the experiments were carried out in strict accordance with the Animal Research: Reporting of *In Vivo* Experiments (ARRIVE) guidelines for the care and use of laboratory animals.

**Data availability.** All RNA-seq data (three independent biological replicates for each sample) have been submitted to the NCBI Sequence Read Archive (SRA) under BioProject accession no. PRJNA907971.

## SUPPLEMENTAL MATERIAL

Supplemental material is available online only.
**SUPPLEMENTAL FILE 1**, PDF file, 1.1 MB.
**SUPPLEMENTAL FILE 2**, XLS file, 0.4 MB.
**SUPPLEMENTAL FILE 3**, XLS file, 0.5 MB.

## ACKNOWLEDGMENTS

This work was supported by the National Natural Science Foundation of China (grant no. 81973983 and 82270015), the Collaborative Tackling and Public Health Collaborative Innovation Project in Anhui Province (no. GXXT-2020-018), the Joint Construction Project of Clinical Medicine University and Hospital (no. 2021lcxk006), the University Natural Science Research Project of Anhui Province (no. KJ2020A0176 and 2022AH051148), the Natural Science Foundation in Anhui Province (no. 2208085MH264), the China Postdoctoral Science Foundation (2022M720196), the Anhui Postdoctoral Science Foundation (no. YJS20210267), and the Project Supported by Anhui Medical University (2021xkj138).

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
