## [Reviewer comments · Microbiology Spectrum]

Microbiology Spectrum

Gallium nitrate enhances antimicrobial activity of colistin against *Klebsiella pneumoniae* by inducing ROS accumulation

Mingjuan Guo, Ping Tian, Qingqing Li, Bao Meng, Yuting Ding, Yan-Yan Liu, Yasheng Li, Liang Yu, and Jiabin Li

Corresponding Author(s): Liang Yu and Jiabin Li, First Affiliated Hospital of Anhui Medical University

Review Timeline:

Submission Date:	January 23, 2023
Editorial Decision:	February 24, 2023
Revision Received:	April 16, 2023
Editorial Decision:	April 27, 2023
Revision Received:	May 5, 2023
Accepted:	May 8, 2023

Editor: Prabakaran Narayanasamy

Reviewer(s): The reviewers have opted to remain anonymous.

Transaction Report:

DOI: <https://doi.org/10.1128/spectrum.00334-23>

February 24, 2023

Prof. Jiabin Li
First Affiliated Hospital of Anhui Medical University
Department of Infectious Diseases
No. 218 Jixi Road, Shushan District, Hefei, Anhui Province, P. R. China
Hefei, Anhui
China

Re: Spectrum00334-23 (Gallium nitrate enhances antimicrobial activity of colistin against *Klebsiella pneumoniae* by inducing ROS accumulation)

Dear Prof. Jiabin Li:

As mentioned by the Reviewers, your article has many concerns to be answered. We recommend you to answer all those comments very carefully in the manuscript for future consideration.

Regards,

Link Not Available

Sincerely,

Prabakaran Narayanasamy

Journals Department
Reviewer comments:

Reviewer #1 (Comments for the Author):

GENERAL COMMENTS

This manuscript reports that the efficacy of colistin and gallium nitrate against *Klebsiella pneumoniae* is enhanced when the two agents are used in combination. Data include in vitro MIC testing and growth curves, as well as in vivo data using a murine lung

infection model. In vitro data include evidence that this enhanced antimicrobial activity extends to colistin resistant strains of *K. pneumoniae*. The authors also provide evidence that this combination therapy is influenced by iron availability, results in enhanced ROS levels in the bacteria (based on use of ROS-sensitive fluorescent probes), and that these enhanced ROS levels are responsible for the enhanced antimicrobial activity based on the ability of an ROS scavenger to mitigate the effect of the combination therapy. The authors also report novel findings that treatment with the two drugs inhibits *K. pneumoniae* expression of SODs and catalase, which would be expected to be induced by ROS, although enzymatic activity was not apparently measured. A decrease in antioxidant activity removing enzymes combined with enhanced ROS production could be the mechanism of the increased antimicrobial activity observed, but the work reported is not of sufficient depth to definitively address this possibility. Strengths of the work include: 1) the potential clinical impact of the work given the global challenge of the growing number of MDR *K. pneumoniae*; 2) the large number of *K. pneumoniae* isolates studied in vitro; 3) the extension of in vitro data (including both planktonic and biofilm growth) to in vivo efficacy; 4) efforts to define mechanism of action; and 5) the unique finding of drug-mediated suppression of SOD and catalase expression. Limitations of the work include: 1) lack of key aspects of methodologic detail as delineated below; 2) reliance on a single ROS scavenger to test role of ROS in activity without consideration of off-target effects; 3) some shortcomings in citing prior literature of relevance; 4) concluding results beyond those supported by the data reported; 5) the lack of data as to whether the results are applicable only to gallium nitrate or other gallium formulations (e.g. gallium maltolate and gallium protoporphyrin) which are also under development as gallium-based antimicrobial agents; and 6) the key finding of the work, synergism of gallium with colistin, has been previously described in other organisms. Although the principal observation of synergism of gallium nitrate with colistin and induction of ROS by each of these agents alone has been previously reported in other Gram-negative pathogens and in the case of ROS production in *K. pneumoniae*, the work does incrementally advance the field by extending the gallium/colistin synergism observation to *K. pneumoniae* and efforts to tie this to ROS levels. Addressing the above limitations could enhance the overall impact of the manuscript.

SPECIFIC COMMENTS:

Major Comments:

1. Line 112: The authors should clarify as to whether these were 1200 truly unique isolates and not multiple isolates obtained from the same patient from different sites or sequentially over a single infection. A summation of the anatomic source of the isolates would also be helpful.
2. Line 132 and line 155: Given the impact of exogenous iron shown by the authors, in order to interpret the MIC data, it would be beneficial to know the iron concentration present in the media used for the in vitro studies - MHA and M9CA.
3. Line 145: on what basis were the 20 isolates selected for PFGE examination?
4. Lines 194-209 and Lines 295-306: The ROS detection experiments were conducted such that the bacteria were first treated with drug and then incubated with the ROS sensitive probe. The authors have not controlled for the possibility that colistin and/or gallium nitrate alter access of the probes to the intracellular space of the bacteria, thus potentially leading to enhanced fluorescence unrelated to differences in ROS levels. Have the authors controlled for this or alternatively loaded the bacteria with the probe prior to adding the antibiotic(s) to see if the results are the same?
5. Lines 307-316: It is difficult to reach a definitive conclusion from the results of a single ROS scavenger, as there may be off-target effects. Although the DMSO effects noticed on antimicrobial activity and ROS detection may well be due to the ROS scavenging ability of DMSO, alternative explanations should be excluded. Regarding ROS detection, does DMSO impact DCF uptake into the bacteria or does it have any ability to quench fluorescence? Regarding antimicrobial activity, could DMSO alter uptake of or binding of colistin or gallium nitrate to *K. pneumoniae*? What is the impact of other ROS scavenging compounds, such as dimethyl thiourea, ascorbic acid, mannitol, etc., on antimicrobial activity? Do they yield the same results as DMSO?
6. Lines 225-242 and 385-397: What *K. pneumoniae* strain was used for the murine studies and what are its MICs to colistin and gallium nitrate? Why was this strain selected for the in vivo studies?
7. line 283 (Figure 3) - why were different concentrations of colistin and gallium nitrate chosen for this sets of experiments than those used for the generation of data shown in Figure 2?
8. Line 287 (Figure 3C): This figure is very difficult to interpret given the overlapping tracings and lack of ability to discern the specific symbols. It should be made clearer. Also, it would be helpful for the authors to indicate why strain ATCC 43816 was chosen for this figure and what are its MICs to colistin and gallium nitrate.
9. Lines 347-348: Gene expression changes do not always correspond to changes in enzymatic activity. Enzymatic activity is the most critical measurement. Therefore, what is the impact of the colistin/gallium nitrate combination on catalase and SOD enzymatic activity. If SOD activity is altered, which of the SODs is impacted.
10. Does the combination of colistin and gallium nitrate have any effect on *K. pneumoniae* capsule formation, which is a critical virulence factor for infection with this organism?

Minor Comments:

1. Line 98 - In addition to reference 16, the authors should cite the work of Goss et al (ref 31) at this point that also demonstrated synergism of gallium nitrate with several conventional antibiotics against *Pseudomonas aeruginosa*, including colistin. This reference should be added.
2. Lines 204-205: The statement that hydrogen peroxide treatment leads to hydroxyl radical formation. It is not clear if the authors intend this as a general statement or specifically relates to prior data with *K. pneumoniae*. A reference here to the statement would be beneficial.
3. Lines 303-304: The authors mention data using another probe, 3'-(p-hydroxyphenyl) fluorescein (HPF). The methodology for

this set of experiments is not included in the ROS detection section of the method section. This should be added. The authors also imply (line 304) that HPF is specific for the detection of hydroxyl radical. However, it will also detect formation of peroxynitrite, which should be noted.

4. Line 322: Catalase and SOD are not "respiratory enzymes", but rather antioxidant enzymes

5. Line 706: The legend of Figure 8 should include the strain of *K. pneumoniae* used, in addition to mentioning it in the methods section where it is currently located.

Reviewer #2 (Comments for the Author):

This work by Guo et al, describe that $Ga(NO_3)_3$ notably enhances the antimicrobial activity of colistin against *K. pneumoniae* in vitro and in vivo, and attempted to explain how $Ga(NO_3)_3$ impacts antibacterial activity via repressing the expression of antioxidant activity genes and increasing the intracellular accumulation of reactive oxygen species (ROS). This is not surprise, but add several important inputs. Thus, it might be accepted after appropriate revision. However, its current version is hampered by several limitations as follows:

1. I would like some more information on the number of repetitions for spot dilution assay, particularly for Fig 1, Fig S2 and Fig S6.

2. Line 166 should read "OD600".

3. Line 721 should read "*K.pn* infected control".

4. In Fig 8D, there are two P vlaues < 0.001, please correct it.

5. In Supplemental Material, there are two Fig S6, please correct it.

6. Lots of key literatures on MCR colistin resistance/ROS production are missing (Xu Y, JBC, 2018; Xu Y mBio, 2018). In addition, reference 32 and 39 are the same, please correct it.

7. Can the authors indicate effects of colistin and GaNt on *Klebsiella pneumoniae* biofilm formation? Because GaNt can prevent *Pseudomonas aeruginosa* biofilm formation (Kaneko, Yukihiro et al, The Journal of clinical investigation 117 4 (2007): 877-88.).

8. Introduction and discussion should be expanded by adding the family of MCR colistin resistance (Zhang et al, Trends Biochem Sci, 2019; Zhang Adv Sci, 2019 a,b).

Reviewer #3 (Comments for the Author):

Klebsiella pneumoniae is a pathogen of clinical concern and there is an urgent need for effective treatment due to its drug resistance. Colistin is an important antimicrobial for the treatment of carbapenem-resistant *K. pneumoniae* infection. Based on their experimental results, the authors believe that GaNt has a significant effect on the antibacterial activity of colistin against *K. pneumoniae* by inducing ROS accumulation, and GaNt has the potential to improve the therapeutic effect of colistin against bacterial infection as a novel adjuvant of colistin. This information is of interest to the audience in the field of infection. However, some issues need to be clarified.

1. Line 85: According to the statement understanding, "GaNt" should be "Ganit". Please check it.

2. Line 131-133: When a volume of 1 mL of double-diluted antimicrobial was added to 9 mL of MHA medium, what concentration of antimicrobial did the authors want to configure? Moreover, the total volume of the medium is only 10 ml, which does not seem to match the volume recommended by CLSI.

3. Line 257: The number of antimicrobials tested described in "Results" does not match what is described in "Method" (Line 131).

4. Only one strain of *Klebsiella pneumoniae* ATCC43816 was tested in ROS assay. Please specify the basis for selecting this strain of bacteria for ROS assay. If you can test a few more strains, the results may be more convincing.

5. RNA sequencing results are somewhat unstable, and it would be better if RT-PCR could be used to verify the differences in gene expression involved in bacterial antioxidant activity.

Staff Comments:

Preparing Revision Guidelines

To submit your modified manuscript, log onto the eJP submission site at <https://spectrum.msubmit.net/cgi-bin/main.plex>. Go to Author Tasks and click the appropriate manuscript title to begin the revision process. The information that you entered when you

first submitted the paper will be displayed. Please update the information as necessary. Here are a few examples of required updates that authors must address:

Please return the manuscript within 60 days; if you cannot complete the modification within this time period, please contact me. If you do not wish to modify the manuscript and prefer to submit it to another journal, please notify me of your decision immediately so that the manuscript may be formally withdrawn from consideration by Microbiology Spectrum.

GENERAL COMMENTS

This manuscript reports that the efficacy of colistin and gallium nitrate against *Klebsiella pneumoniae* is enhanced when the two agents are used in combination. Data include *in vitro* MIC testing and growth curves, as well as *in vivo* data using a murine lung infection model. *In vitro* data include evidence that this enhanced antimicrobial activity extends to colistin resistant strains of *K. pneumoniae*. The authors also provide evidence that this combination therapy is influenced by iron availability, results in enhanced ROS levels in the bacteria (based on use of ROS-sensitive fluorescent probes), and that these enhanced ROS levels are responsible for the enhanced antimicrobial activity based on the ability of an ROS scavenger to mitigate the effect of the combination therapy. The authors also report novel findings that treatment with the two drugs inhibits *K. pneumoniae* expression of SODs and catalase, which would be expected to be induced by ROS, although enzymatic activity was not apparently measured. A decrease in antioxidant activity removing enzymes combined with enhanced ROS production could be the mechanism of the increased antimicrobial activity observed, but the work reported is not of sufficient depth to definitively address this possibility. Strengths of the work include: 1) the potential clinical impact of the work given the global challenge of the growing number of MDR *K. pneumoniae*; 2) the large number of *K. pneumoniae* isolates studied *in vitro*; 3) the extension of *in vitro* data (including both planktonic and biofilm growth) to *in vivo* efficacy; 4) efforts to define mechanism of action; and 5) the unique finding of drug-mediated suppression of SOD and catalase expression. Limitations of the work include: 1) lack of key aspects of methodologic detail as delineated below; 2) reliance on a single ROS scavenger to test role of ROS in activity without consideration of off-target effects; 3) some shortcomings in citing prior literature of relevance; 4) concluding results beyond those supported by the data reported; 5) the lack of data as to whether the results are applicable only to gallium nitrate or other gallium formulations (e.g. gallium maltolate and gallium protoporphyrin) which are also under development as gallium-based antimicrobial agents; and 6) the key finding of the work, synergism of gallium with colistin, has been previously described in other organisms. Although the principal observation of synergism of gallium nitrate with colistin and induction of ROS by each of these agents alone has been previously reported in other Gram-negative pathogens and in the case of ROS production in *K. pneumoniae*, the work does incrementally advance the field by extending the gallium/colistin synergism observation to *K. pneumoniae* and efforts to tie this to ROS levels. Addressing the above limitations could enhance the overall impact of the manuscript.

SPECIFIC COMMENTS:

Major Comments:

1. Line 112: The authors should clarify as to whether these were 1200 truly unique isolates and not multiple isolates obtained from the same patient from different sites or sequentially over a single infection. A summation of the anatomic source of the isolates would also be helpful.
2. Line 132 and line 155: Given the impact of exogenous iron shown by the authors, in order to interpret the MIC data, it would be beneficial to know the iron concentration present in the media used for the *in vitro* studies – MHA and M9CA.
3. Line 145: on what basis were the 20 isolates selected for PFGE examination?

4. Lines 194-209 and Lines 295-306: The ROS detection experiments were conducted such that the bacteria were first treated with drug and then incubated with the ROS sensitive probe. The authors have not controlled for the possibility that colistin and/or gallium nitrate alter access of the probes to the intracellular space of the bacteria, thus potentially leading to enhanced fluorescence unrelated to differences in ROS levels. Have the authors controlled for this or alternatively loaded the bacteria with the probe prior to adding the antibiotic(s) to see if the results are the same?
5. Lines 307-316: It is difficult to reach a definitive conclusion from the results of a single ROS scavenger, as there may be off-target effects. Although the DMSO effects noticed on antimicrobial activity and ROS detection may well be due to the ROS scavenging ability of DMSO, alternative explanations should be excluded. Regarding ROS detection, does DMSO impact DCF uptake into the bacteria or does it have any ability to quench fluorescence? Regarding antimicrobial activity, could DMSO alter uptake of or binding of colistin or gallium nitrate to *K. pneumoniae*? What is the impact of other ROS scavenging compounds, such as dimethyl thiourea, ascorbic acid, mannitol, etc., on antimicrobial activity? Do they yield the same results as DMSO?
6. Lines 225-242 and 385-397: What *K. pneumoniae* strain was used for the murine studies and what are its MICs to colistin and gallium nitrate? Why was this strain selected for the *in vivo* studies?
7. line 283 (Figure 3) – why were different concentrations of colistin and gallium nitrate chosen for this sets of experiments than those used for the generation of data shown in Figure 2?
8. Line 287 (Figure 3C): This figure is very difficult to interpret given the overlapping tracings and lack of ability to discern the specific symbols. It should be made clearer. Also, it would be helpful for the authors to indicate why strain ATCC 43816 was chosen for this figure and what are its MICs to colistin and gallium nitrate.
9. Lines 347-348: Gene expression changes do not always correspond to changes in enzymatic activity. Enzymatic activity is the most critical measurement. Therefore, what is the impact of the colistin/gallium nitrate combination on catalase and SOD enzymatic activity. If SOD activity is altered, which of the SODs is impacted.
10. Does the combination of colistin and gallium nitrate have any effect on *K. pneumoniae* capsule formation, which is a critical virulence factor for infection with this organism?

Minor Comments:

1. Line 98 – In addition to reference 16, the authors should cite the work of Goss et al (ref 31) at this point that also demonstrated synergism of gallium nitrate with several conventional antibiotics against *Pseudomonas aeruginosa*, including colistin. This reference should be added.
2. Lines 204-205: The statement that hydrogen peroxide treatment leads to hydroxyl radical formation. It is not clear if the authors intend this as a general statement or specifically relates to prior data with *K. pneumoniae*. A reference here to the statement would be beneficial.
3. Lines 303-304: The authors mention data using another probe, 3'-(p-hydroxyphenyl) fluorescein (HPF). The methodology for this set of experiments is not included in the ROS detection section of the method section. This should be added. The authors also imply (line 304) that HPF is specific for the

detection of hydroxyl radical. However, it will also detect formation of peroxynitrite, which should be noted.

4. Line 322: Catalase and SOD are not “respiratory enzymes”, but rather antioxidant enzymes

5. Line 706: The legend of Figure 8 should include the strain of *K. pneumoniae* used, in addition to mentioning it in the methods section where it is currently located.

Reviewer comments:

Reviewer #1 (Comments for the Author):

GENERAL COMMENTS

This manuscript reports that the efficacy of colistin and gallium nitrate against *Klebsiella pneumoniae* is enhanced when the two agents are used in combination. Data include in vitro MIC testing and growth curves, as well as in vivo data using a murine lung infection model. In vitro data include evidence that this enhanced antimicrobial activity extends to colistin resistant strains of *K. pneumoniae*. The authors also provide evidence that this combination therapy is influenced by iron availability, results in enhanced ROS levels in the bacteria (based on use of ROS-sensitive fluorescent probes), and that these enhanced ROS levels are responsible for the enhanced antimicrobial activity based on the ability of an ROS scavenger to mitigate the effect of the combination therapy. The authors also report novel findings that treatment with the two drugs inhibits *K. pneumoniae* expression of SODs and catalase, which would be expected to be induced by ROS, although enzymatic activity was not apparently measured. A decrease in antioxidant activity removing enzymes combined with enhanced ROS production could be the mechanism of the increased antimicrobial activity observed, but the work reported is not of sufficient depth to definitively address this possibility. Strengths of the work include: 1) the potential clinical impact of the work given the global challenge of the growing number of MDR *K. pneumoniae*; 2) the large number of *K. pneumoniae* isolates studied in vitro; 3) the extension of in vitro data (including both planktonic and biofilm growth) to in vivo efficacy; 4) efforts to define mechanism of action; and 5) the unique finding of drug-mediated suppression of SOD and catalase expression. Limitations of the work include: 1) lack of key aspects of methodologic detail as delineated below; 2) reliance on a single ROS scavenger to test role of ROS in activity without consideration of off-target effects; 3) some shortcomings in citing prior literature of relevance; 4) concluding results beyond those supported by the data

reported; 5) the lack of data as to whether the results are applicable only to gallium nitrate or other gallium formulations (e.g. gallium maltolate and gallium protoporphyrin) which are also under development as gallium-based antimicrobial agents; and 6) the key finding of the work, synergism of gallium with colistin, has been previously described in other organisms. Although the principal observation of synergism of gallium nitrate with colistin and induction of ROS by each of these agents alone has been previously reported in other Gram-negative pathogens and in the case of ROS production in *K. pneumoniae*, the work does incrementally advance the field by extending the gallium/colistin synergism observation to *K. pneumoniae* and efforts to tie this to ROS levels. Addressing the above limitations could enhance the overall impact of the manuscript.

Response: Thank you for your very critical comments and helpful suggestions. We have carefully revised the manuscript and performed additional experiments according to your comments. Below we provide responses for each point.

SPECIFIC COMMENTS:

Major Comments:

1. Line 112: The authors should clarify as to whether these were 1200 truly unique isolates and not multiple isolates obtained from the same patient from different sites or sequentially over a single infection. A summation of the anatomic source of the isolates would also be helpful.

Response: Thank you for pointing this out. According to the data from 59 member units of the Anhui Center for Surveillance of Bacterial Resistance (Liu Y, *et al. Infection and Drug Resistance* (2022): 7537-7553.), these strains were unique isolates. We also rephrased the statement as following (Line 113-115): "... 1200 nonrepetitive *K. pneumoniae* strains were isolated from clinical samples of inpatients and outpatients from the Anhui Center for Surveillance of Bacterial Resistance with qualified data."

2. Line 132 and line 155: Given the impact of exogenous iron shown by the authors,

in order to interpret the MIC data, it would be beneficial to know the iron concentration present in the media used for the in vitro studies - MHA and M9CA.

Response: Thanks for raising an important question to us. By following your comments, we added the information about the ferric ions concentrations into the revised manuscript: "First, 1 mL of the serial dilutions of extracted antibiotics was added to 19 mL of still liquid MHA medium containing < 2.2 µg/mL iron (R Girardello, *et al.* Journal of Clinical Microbiology 50.7 (2012): 2414-2418.) at a temperature of 45–50 °C under aseptic conditions " (line 133-135) and "M9CA medium (Sangon Biotech, China) containing 0.07 µg/mL iron determined by ICP-OES at Shiyanjia lab (www.Shiyanjia.com), was used as a growth medium for cultivation of bacteria in this assay." (line 161-164)

3. Line 145: on what basis were the 20 isolates selected for PFGE examination?

Response: Sorry for the unclear description and we have clarified this point. We have revised the sentence, "10 COL-resistant ($MIC \geq 8$ µg/mL) and 10 COL-sensitive strains ($MIC \leq 1$ µg/mL) (see details in Table S1) were selected from *K. pneumoniae* strains described above and" (line 150-152).

4. Lines 194-209 and Lines 295-306: The ROS detection experiments were conducted such that the bacteria were first treated with drug and then incubated with the ROS sensitive probe. The authors have not controlled for the possibility that colistin and/or gallium nitrate alter access of the probes to the intracellular space of the bacteria, thus potentially leading to enhanced fluorescence unrelated to differences in ROS levels. Have the authors controlled for this or alternatively loaded the bacteria with the probe prior to adding the antibiotic(s) to see if the results are the same?

Response: We appreciate your very careful reading and constructive recommendation. By following your comment, the bacteria were incubated with the

ROS sensitive probe after treated with GaNt and colistin and the results were showed in Fig. SR-11(as following). In line with the results of prior analyses, we found that a significant increase in ROS accumulation caused by GaNt-colistin combination, but only a modest increase induced GaNt treatment, similar to colistin treatment.

Fig. SR-11. Drug-treatment increased ROS accumulation. Before (A) or after (B) samples were incubated with 10 mM carboxy-H₂DCFDA for 30 min and washed twice, *K. pneumoniae* ATCC 43816 were untreated (Control) or treated with 2 µg/mL colistin (COL), 20 µg/mL GaNt (Ga³⁺), 2 µg/mL colistin and 20 µg/mL GaNt (COL&Ga³⁺), and 0.1% H₂O₂ (H₂O₂), respectively for 2 h. Reactive oxygen species (ROS) were determined using flow cytometry.

5. Lines 307-316: It is difficult to reach a definitive conclusion from the results of a single ROS scavenger, as there may be off-target effects. Although the DMSO effects noticed on antimicrobial activity and ROS detection may well be due to the ROS scavenging ability of DMSO, alternative explanations should be excluded. Regarding ROS detection, does DMSO impact DCF uptake into the bacteria or does it have any ability to quench fluorescence? Regarding antimicrobial activity, could DMSO alter uptake of or binding of colistin or gallium nitrate to *K. pneumoniae*? What is the impact of other ROS scavenging compounds, such as dimethyl thiourea, ascorbic acid, mannitol, etc., on antimicrobial activity? Do they yield the same results as DMSO?

Response: We appreciate the concern. To to exclude DMSO with undesirable side

effects, other ROS scavenging compounds (dimethyl thiourea and ascorbic acid) were tested to see if they would suppress the antibacterial effect of GaNt-colistin and ROS accumulation. We found that when exogenous dimethyl thiourea or ascorbic acid was added to bacterial cultures containing GaNt and colistin, the intracellular ROS accumulation was reduced and bacterial clearance was decreased (Fig. S6).

By following your comment, we have carefully revised the manuscript accordingly as following: "Additionally, similar phenomena were observed when other exogenous ROS scavenging compounds (dimethyl thiourea and ascorbic acid) were added to the cultures (Fig. S6)." (line 342-344).

In addition, to make the above results more convincing, another two *Klebsiella pneumoniae* strains (ATCC 700603 and GN 182201) were chosen for rapid killing and ROS assay. These new data were presented in the Fig. S7 in the revised manuscript, and we also have added sentences into the results section as following (line 347-351): "To further validate these results, we performed rapid killing and ROS assay using another two *Klebsiella pneumoniae* strains (ATCC 700603 and GN 182201). Similar to ATCC 43816, ATCC 700603 and GN 182201 were protected by exogenous DMSO or ascorbic acid from GaNt-colistin-mediated bacterial killing and intracellular ROS accumulation (Fig. S7).".

6. Lines 225-242 and 385-397: What *K. pneumoniae* strain was used for the murine studies and what are its MICs to colistin and gallium nitrate? Why was this strain selected for the *in vivo* studies?

Response: Thank you for pointing out the unclear description. **1)** *Klebsiella pneumoniae* ATCC 43816 exhibited a colistin MIC of 0.5 µg/mL (see Table S1) and were resistance to gallium nitrate (MIC = 100 µg/mL). **2)** *K. pneumoniae* ATCC 43816 was selected for *in vivo* and *vitro* study because it is a well-studied strain, capable of causing a respiratory disease in mouse models (Silver, Rebecca J., *et al.* Antimicrobial Agents and Chemotherapy 63.8 (2019): e02674-18; Walker, Kimberly A., *et al.* MBio 10.2 (2019): e00089-19; Fodah, Ramy A., *et al.* PloS one 9.9 (2014): e107394.). ATCC 43816 was also selected for experiments *in vivo* in our team (Wu, Ting, *et al.*

Msystems 5.6 (2020): e00587-20; Wu, Ting, *et al.* *Frontiers in Immunology* 11 (2020): 1331.).

Given these, we modified the statement in a more appropriate way in the revised manuscript: “*K. pneumoniae* ATCC 43816, as a well-studied strain capable of causing a respiratory disease in mouse models (Silver, Rebecca J., *et al.* *Antimicrobial Agents and Chemotherapy* 63.8 (2019): e02674-18; Walker, Kimberly A., *et al.* *MBio* 10.2 (2019): e00089-19; Fodah, Ramy A., *et al.* *PloS one* 9.9 (2014): e107394.), was chosen for further studies *in vivo* and *in vitro*. Consistent with spot dilution assay, rapid killing (Fig. 3B) and bacterial growth curve assay (Fig. 3C) also showed that GaNt addition markedly decreased the growth of *K. pneumoniae* ATCC 43816. We also performed”(line 313-317), “... inoculated intranasally with 1×10^6 CFU of *K. pneumoniae* ATCC 43816 ” (line 252) and “ Figure 8. Effect of colistin and GaNt on *Klebsiella pneumoniae* ATCC 43816 induced lung injury. ”(line 793).

7. line 283 (Figure 3) - why were different concentrations of colistin and gallium nitrate chosen for this sets of experiments than those used for the generation of data shown in Figure 2?

Response: We highly appreciate the careful reading. In Fig 2, the 10 colistin -resistant MDR *K. pneumoniae* clinical strains were chosen for spot dilution assay that lead to the concentrations of colistin and gallium nitrate used was higher than 10 colistin colistin-sensitive clinical strains (Fig S2) and the strains used in Fig 3.

In addition, with **gallium maltolate replacing GaNT**, we performed spot dilution assay as as decribed in Methods, and our results showed gallium maltolate also enhanced the growth inhibitory effect of colistin on *K. pneumoniae* (decreased by 2–3 log₁₀ CFU/mL) (Fig. S4). These new data are presented in the revised Fig. S4 and we have rephrased the statement as following (Line 309-310): “A similar phenomenon was observed when polymyxin B was used instead of colistin (Fig. S3) or gallium maltolate replaced GaNT (Fig. S4).

8. Line 287 (Figure 3C): This figure is very difficult to interpret given the overlapping tracings and lack of ability to discern the specific symbols. It should be made clearer. Also, it would be helpful for the authors to indicate why strain ATCC 43816 was chosen for this figure and what are its MICs to colistin and gallium nitrate.

Response: We are so sorry for the unclear presentation. We have carefully revised the figure (Fig. 3C) and the statements (line 313-314) according to your comments in order to make them clear.

9. Lines 347-348: Gene expression changes do not always correspond to changes in enzymatic activity. Enzymatic activity is the most critical measurement. Therefore, what is the impact of the colistin/gallium nitrate combination on catalase and SOD enzymatic activity. If SOD activity is altered, which of the SODs is impacted.

Response: We do apologize for an inaccurate conclusion here and changed the expression in this section as “These observations support the notion that genes involved in antioxidant activity of *K. pneumoniae* could be suppressed by GaNt with or without colistin, which appears to cause intracellular ROS accumulation.”(line 383-385). In addition, we found the reversal of the repression effect of gallium nitrate by nitric oxide (NO). The studies of the effect of the gallium nitrate with or without colistin/NO donor on catalase and SOD enzymatic activity are undergoing in our laboratory and the results will be presented in a separated manuscript.

10. Does the combination of colistin and gallium nitrate have any effect on *K. pneumoniae* capsule formation, which is a critical virulence factor for infection with this organism?

Response: Thank you for your advice. We measured the Capsular polysaccharide production of *K. pneumoniae* as previously described (Ma, Xuejiao et al, Infect Drug Resist. 2022;15:3513-3522.). Briefly, 500 μ L of overnight broth-cultured bacteria was mixed with 100 μ L of 1% Zwittergent 3–14 (Sigma-Aldrich, Milwaukee, WI) in 100

mM citric acid (pH 2.0) and then incubated at 50°C for 20 min. After centrifugation, 250 µL of the supernatant was transferred and added with 1 mL of cold ethanol. The mixture was incubated at 4°C for 20 min for precipitation. After centrifugation, the pellet was dried and dissolved in 200 µL of distilled water, and then 1200 µL of 12.5 mM tetraborate in concentrated H₂SO₄ was added. After vigorous vortex, the mixture was boiled for 5 min. After cooling, 20 µL of 0.15% 3-hydroxydiphenol (Sigma-Aldrich) was added. Then, the absorbance at 520 nm was measured. Our result showed that there was no statistical difference in Fig. SR-12.

Fig. SR-12. Quantitative Measurement of Bacterial Capsular Polysaccharide Capsule production of *K. pneumoniae* ATCC 43816 were assessed by measurement of absorbance at 520 nm. According to Fig 3C, at colistin concentration of 0.5 µg/mL, GaNt (25 µg/mL) plus colistin and colistin alone both do not influence bacterial growth. Student's two-tailed t-test was performed to determine statistically no differences between samples. The amount of CPS was represented by OD₅₂₀ of four independent experiments.

Minor Comments:

1. Line 98 - In addition to reference 16, the authors should cite the work of Goss et al (ref 31) at this point that also demonstrated synergism of gallium nitrate with several conventional antibiotics against *Pseudomonas aeruginosa*, including colistin. This reference should be added.

Response: Thank you for your advice. We corrected it by citing the mentioned article (Goss, Christopher H., et al. *Science translational medicine* 10.460 (2018): eaat7520.) here.

2. Lines 204-205: The statement that hydrogen peroxide treatment leads to hydroxyl radical formation. It is not clear if the authors intend this as a general statement or specifically relates to prior data with *K. pneumoniae*. A reference here to the statement would be beneficial.

Response: We agree. Following your suggestion, we cited an article in which hydrogen peroxide treatment leads to hydroxyl radical formation (Ikai H, et al. *Antimicrob Agents Chemother.* 2010;54(12):5086-5091) in line 217.

3. Lines 303-304: The authors mention data using another probe, 3',3'-bis(carboxymethyl)-6-(p-hydroxyphenyl) fluorescein (HPF). The methodology for this set of experiments is not included in the ROS detection section of the method section. This should be added. The authors also imply (line 304) that HPF is specific for the detection of hydroxyl radical. However, it will also detect formation of peroxynitrite, which should be noted.

Response: We have clarified this point in the revised manuscript. Following your suggestion, we have revised the statements in the Method section:

"ROS and Hydroxyl Radical assay.

To assess intracellular ROS accumulation, the fluorescent probe carboxy-H₂DCFDA (Thermo Fisher, Waltham, MA, USA) was utilized. Since peroxynitrite was absent in the medium as *K. pneumoniae* was not exposed to acidified nitrite, the possibility of the formation of hydroxyl radical from peroxynitrite was ruled out (Setsukinai K, et al. *J Biol Chem.* 2003;278(5):3170-3175.). HPF was used for the detection of hydroxyl radical, although it gets oxidized by hydroxyl radical and peroxynitrite (Paul Avraneel, et al. *Antimicrobial agents and chemotherapy* vol. 66,5 (2022): e0228521.). "(line 203-208).

4. Line 322: Catalase and SOD are not "respiratory enzymes", but rather antioxidant enzymes

Response: Thank you for providing this significant insight. By following your comments, we change the words to "antioxidant enzymes" for better understanding.

5. Line 706: The legend of Figure 8 should include the strain of *K. pneumoniae* used, in addition to mentioning it in the methods section where it is currently located.

Response: We are very grateful to the reviewer for pointing this out. We corrected in the revised manuscript (Seen in line 793).

Reviewer #2 (Comments for the Author):

This work by Guo et al, describe that $\text{Ga}(\text{NO}_3)_3$ notably enhances the antimicrobial activity of colistin against *K. pneumoniae* in vitro and in vivo, and attempted to explain how $\text{Ga}(\text{NO}_3)_3$ impacts antibacterial activity via repressing the expression of antioxidant activity genes and increasing the intracellular accumulation of reactive oxygen species (ROS). This is not surprise, but add several important inputs. Thus, it might be be accepted after appropriate revision. However, its current version is hampered by several limitations as follows:

Response: Thank you so much for your assessment of our manuscript. We have carefully revised the manuscript according to your comments.

1. I would like some more information on the number of repetitions for spot dilution assay, particularly for Fig 1, Fig S2 and Fig S6.

Response: By following your comments, We added the number of repetitions used for spot dilution assay. Please see details in the figure legends (i.e., Fig 1, Fig 2, Fig S2, Fig S6 and Fig S7).

2. Line 166 should read "OD600".

Response: We corrected it.

3. Line 721 should read "K.pn infected control".

Response: We corrected it.

4. In Fig 8D, there are two P vlaues < 0.001, please correct it.

Response: We corrected it. We deleted the wrong *P* vlaue (PBS versus *K.pn* infected control) in the figure Fig 8D.

5. In Supplemental Material, there are two Fig S6, please correct it.

Response: We corrected it.

6. Lots of key literatures on MCR colistin resistance/ROS production are missing (Xu Y, JBC, 2018; Xu Y mBio, 2018). In addition, reference 32 and 39 are the same, please correct it.

Response: We agree. By following your comments, we added the following sentences into the revised manuscript (line 438-440): "but the expression of mobile colistin resistance (MCR) enzymes (e.g., EptA, MCR-1, MCR-2, etc.) or nonmobile colistin resistance (NMCR) enzymes (e.g., NMCR-1, NMCR-1.2, NMCR-1.8, etc.) is shown to remodel the surface of enteric bacteria (Xu Y mBio, 2018,Zhang Adv Sci, 2019 a,b)," and changed "*K. pneumoniae* strains" to "Gram-negative bacteria" (line 441) .

7. Can the authors indicate effects of colistin and GaNt on Klebsiella pneumoniae biofilm formation? Because GaNt can prevent Pseudomonas aeruginosa biofilm formation (Kaneko, Yukihiro et al, The Journal of clinical investigation 117 4 (2007): 877-88.).

Response: Thank you for your advice. We measured the biofilm formation of *K. pneumoniae* using crystal violet as previously described (Chen, Haoran et al, Front Cell Infect Microbiol. 2022;12:927289.). Briefly, overnight cultures were standardized to exponential phase OD600 ~0.3, and then diluted (1:100) with fresh broth. Samples

(200 μ L) were loaded into individual wells of 96-well plates and incubated at 37°C for 24 h. The medium was discarded, and the wells were washed three times with PBS. Next, the bacterial biofilms were fixed with 95% formalin for 15 min and stained with 0.1% crystal violet solution for 20 min. Then, the excess stain was removed by washing twice with PBS, the stained biofilms were dried for 1 h and extracted with 33% glacial acetic acid. The amount of biofilm produced was quantified by measuring the optical density at 590 nm using a Tecan Spark Multimode Microplate Reader. Our result showed that there was no statistical difference in Fig. SR-21.

Fig. SR-21. Crystal violet staining of biofilms under different treatments. Biofilm formation of *K. pneumoniae* ATCC 43816 were assessed by measurement of absorbance at 590 nm. According to Fig 3C, at colistin concentration of 0.5 μ g/mL, GaNt (25 μ g/mL) plus colistin and colistin alone both do not influence bacterial growth. Student's two-tailed t-test was performed to determine statistically no differences between samples. The amount of biofilms was represented by OD₅₉₀ of four independent experiments.

8. Introduction and discussion should be expanded by adding the family of MCR colistin resistance (Zhang et al, Trends Biochem Sci, 2019; Zhang Adv Sci, 2019 a,b).
Response: Thanks for your advice. By following your comments, we have revised the sentence in the Introduction section, " the increasing use of polymyxins over the last decade has fueled the generation of polymyxin-resistant *K. pneumoniae* strains

(Rodríguez-Santiago et al, Int J Antimicrob Agents, 2021). ”, as “transferable polymyxin resistance mediated by mobile colistin resistance (MCR) enzymes fueled the generation of polymyxin-resistant bacteria, including polymyxin-resistant *K. pneumoniae* (Zhang et al, Trends Biochem Sci, 2019; Xu Y, JBC, 2018; Rodríguez-Santiago et al, Int J Antimicrob Agents, 2021). ” (line 79-81)

Reviewer #3 (Comments for the Author):

Klebsiella pneumoniae is a pathogen of clinical concern and there is an urgent need for effective treatment due to its drug resistance. Colistin is an important antimicrobial for the treatment of carbapenem-resistant *K. pneumoniae* infection. Based on their experimental results, the authors believe that GaNt has a significant effect on the antibacterial activity of colistin against *K. pneumoniae* by inducing ROS accumulation, and GaNt has the potential to improve the therapeutic effect of colistin against bacterial infection as a novel adjuvant of colistin. This information is of interest to the audience in the field of infection. However, some issues need to be clarified.

Response: Thanks for your positive assessment of our manuscript. Below we provided responses for each point.

1. Line 85: According to the statement understanding, "GaNt" should be "Ganit". Please check it.

Response: We apologize for this omission and corrected it. By following your comments, we changed the “GaNt” to “Ganit, the FDA-approved GaNt formulation,” in this revised manuscript (line 86) and deleted the statement, “the FDA-approved GaNt formulation”(line 94)

2. Line 131-133: When a volume of 1 mL of double-diluted antimicrobial was added to 9 mL of MHA medium, what concentration of antimicrobial did the authors want

to configure? Moreover, the total volume of the medium is only 10 ml, which does not seem to match the volume recommended by CLSI.

Response: Sorry for the inappropriate description and we have clarified this point. By following your comments, we modified the statement about determination of MIC into the Materials and Methods section: “ 24 different antibiotics (see details in Table S1) were used in the agar dilution method. First, 1 mL of the serial dilutions of extracted antibiotics was added to 19 mL of still-liquid MHA medium containing < 2.2 mg/L iron at a temperature of 45–50 °C under aseptic conditions, to perform growth tests of the analyzed strains. The supplemented mediums were homogenized and poured on Petri dishes with a diameter of 9 cm. Then a 0.5 McFarland (McF) suspension was diluted 1:10 in 1 × Phosphate-Buffered Saline (PBS) ”(line 133-137). We also added the information about the concentration of antimicrobial in Table S1.

3. Line 257: The number of antimicrobials tested described in "Results" does not match what is described in "Method" (Line 131).

Response: We appreciate your comment and apologize for our carelessness. We corrected the number in Method(line 133).

4. Only one strain of *Klebsiella pneumoniae* ATCC 43816 was tested in ROS assay. Please specify the basis for selecting this strain of bacteria for ROS assay. If you can test a few more strains, the results may be more convincing.

Response: Thank you for providing this significant insight. By following your comment, another two *Klebsiella pneumoniae* strains (ATCC 700603 and GN 182201) and two ROS scavenging compounds (DMSO and ascorbic acid) were used in rapid killing and ROS assay. The results showed that exogenous 2% DMSO or 5 mM ascorbic acid diminished GaNt-colistin-mediated rapid bacterial killing and reduced the intracellular ROS accumulation in ATCC 700603 and GN 182201.

These new data were presented in the Fig. S7 in the revised manuscript, and we

also have added sentences into the results section as following (line 347-351):“To further validate these results, we performed rapid killing and ROS assay using another two *Klebsiella pneumoniae* strains (ATCC 700603 and GN 182201). Similar to ATCC 43816, ATCC 700603 and GN 182201 were protected by exogenous DMSO or ascorbic acid from GaNt-colistin-mediated bacterial killing and intracellular ROS accumulation (Fig. S7). ”.

5. RNA sequencing results are somewhat unstable, and it would be better if RT-PCR could be used to verify the differences in gene expression involved in bacterial antioxidant activity.

Response: We thank the reviewer for this constructive suggestion. By following your comments, we performed RT-PCR as your suggestion, and the results were showed in Fig. S5. We added sentences into the Methods section as following (line 237-246):

“Quantitative real-time PCR (qRT-PCR) analysis Bacterial gene expression analysis was performed by qRT-PCR as described previously^[33]. The total DNase-treated RNA was extracted using an RNeasy kit (Qiagen, Hilden, Germany) and reversely transcribed to synthesize cDNA using a PrimeScript cDNA synthesis kit (TaKaRa, Kyoto, Japan) according to the manufacturer’ s recommendation. qRT-PCR of *sodA*, *sodB*, *sodC*, *katE* and *katG* was performed using PrimeScript RT master mix (TaKaRa) by a three-step real-time PCR system (Light Cycler 96; Roche, Basel, Switzerland). The amplicon of 16S rRNA was used as an internal control. Primers for qRT-PCR are listed in Table S2. The expression levels of target genes of interest were determined by the $2^{-\Delta\Delta CT}$ calculation method and reported as fold-change values. Reactions were run in triplicate.” and a following sentence to the Results section (line 381-383): “A similar result was obtained when the expression of genes (*sodA/B/C* and *katE/G*) encoding antioxidant enzymes was examined by quantitative reverse transcription PCR (qRT-PCR) (Fig. S8).” .

April 27, 2023

Prof. Jiabin Li
First Affiliated Hospital of Anhui Medical University
Department of Infectious Diseases
No. 218 Jixi Road, Shushan District, Hefei, Anhui Province, P. R. China
Hefei, Anhui
China

Re: Spectrum00334-23R1 (Gallium nitrate enhances antimicrobial activity of colistin against *Klebsiella pneumoniae* by inducing ROS accumulation)

Dear Prof. Jiabin Li:

As this is the last chance to fix the errors, A minor modification is recommended, as per the reviewers comments.

Link Not Available

Sincerely,

Prabakaran Narayanasamy

Journals Department
Reviewer comments:

Reviewer #1 (Comments for the Author):

The authors have addressed my prior comments and concerns as well as those of the other reviewers

Reviewer #2 (Comments for the Author):

It is greatly improved! References should be formatted accordingly.

Reviewer #3 (Comments for the Author):

I mean gallium nitrate is the anhydrate salt of the naturally occurring heavy metal, not naturally occurring metal. Please modify the statement (Line 85) as appropriate.

Staff Comments:

Preparing Revision Guidelines

Please return the manuscript within 60 days; if you cannot complete the modification within this time period, please contact me. If you do not wish to modify the manuscript and prefer to submit it to another journal, please notify me of your decision immediately so that the manuscript may be formally withdrawn from consideration by Microbiology Spectrum.

Reviewer comments:

Reviewer #1 (Comments for the Author):

The authors have addressed my prior comments and concerns as well as those of the other reviewers

Response: Thanks to the reviewer for the positive affirmation!

Reviewer #2 (Comments for the Author):

It is greatly improved! References should be formatted accordingly.

Response: We are very grateful to you for your positive assessment of our manuscript. We also reformatted the references and other parts of the article in accordance with the journal's requirements.

Reviewer #3 (Comments for the Author):

I mean gallium nitrate is the anhydrate salt of the naturally occurring heavy metal, not naturally occurring metal. Please modify the statement (Line 85) as appropriate.

Response: Sorry for the inappropriate description and we have clarified this point. By following your comment, we have re-written the expression as following: "Gallium is a naturally occurring metal with an ionic radius that is nearly identical to that of iron but is redox-inactive. Because gallium is a ferric iron mimetic, gallium compounds was" (line 85-88) and "Therefore, GaNt , the FDA-approved GaNt formulation, represents" (line 94).

May 8, 2023

Prof. Jiabin Li
First Affiliated Hospital of Anhui Medical University
Department of Infectious Diseases
No. 218 Jixi Road, Shushan District, Hefei, Anhui Province, P. R. China
Hefei, Anhui
China

Re: Spectrum00334-23R2 (Gallium nitrate enhances antimicrobial activity of colistin against *Klebsiella pneumoniae* by inducing ROS accumulation)

Dear Prof. Jiabin Li:

Authors addressed all the critiques from reviewers.

Your manuscript has been accepted, and I am forwarding it to the ASM Journals Department for publication. You will be notified when your proofs are ready to be viewed.

Sincerely,

Prabakaran Narayanasamy
Editor, Microbiology Spectrum
